# R2RGEN: REAL-TO-REAL 3D DATA GENERATION FOR SPATIALLY GENERALIZED MANIPULATION

## ABSTRACT

Towards the aim of generalized robotic manipulation, spatial generalization is the most fundamental capability that requires the policy to work robustly under different spatial distribution of objects, environment and agent itself. To achieve this, substantial human demonstrations need to be collected to cover different spatial configurations for training a generalized visuomotor policy via imitation learning. Prior works explore a promising direction that leverages data generation to acquire abundant spatially diverse data from minimal source demonstrations. However, most approaches face significant sim-to-real gap and are often limited to constrained settings, such as fixed-base scenarios and predefined camera viewpoints. In this paper, we propose a real-to-real 3D data generation framework (R2RGen) that directly augments the pointcloud observation-action pairs to generate real-world data. R2RGen is simulator- and rendering-free, thus being efficient and plug-and-play. Specifically, given a single source demonstration, we introduce an annotation mechanism for fine-grained parsing of scene and trajectory. A group-wise augmentation strategy is proposed to handle complex multi-object compositions and diverse task constraints. We further present camera-aware processing to align the distribution of generated data with real-world 3D sensor. Empirically, R2RGen substantially enhances data efficiency on extensive experiments and demonstrates strong potential for scaling and application on mobile manipulation. Website.

## 1 INTRODUCTION

Robotic manipulation with visuomotor policy Chi et al. (2023); Zhao et al. (2023); Fu et al. (2024) has achieved great progress in recent years, while the reliance on large amount of human-collected data during imitation learning becomes the main bottlenecks for application and further scaling up Lin et al. (2024). Unlike most prior work focused on fixed-tabletop arms, this paper studies a more general manipulation setting involving mobile manipulators. Since the mobile base may be located at arbitrary positions, the resulting viewpoint variation further increases the policy's reliance on extensive training data Tan et al. (2024).

***Spatial generalization*** constitutes the primary factor driving the substantial data demand during visuomotor policy learning. As pointed out by Garrett et al. (2024), control difficulty is not uniformly distributed among the human-collected trajectory. Note a trajectory can be divided into two categories of segments: contact-rich segments involving the interaction between robotic arm and objects, and other segments simply indicating the movement of robotic arm in free space, which are also known as *skill* and *motion* segments respectively. Skill segments are generally more challenging, whereas motion segments can often be handled effectively through motion planning. However, even thought skill segments are more informative, the majority of human demonstration effort is typically devoted to teaching motion behaviors. For instance, in a task such as "put apple on plate"—even with identical apple, plate, and pick and place skill—hundreds of demonstrations may be needed to cover varying objects' spatial arrangements and robot's base positions to learn a generalized policy. Therefore, spatial generalization remains a fundamental bottleneck in data efficiency.

To reduce redundant human effort on ensuring spatial generalization, MimicGen Jiang et al. (2024) and follow-up works Hoque et al. (2024); Garrett et al. (2024); Jiang et al. (2024) replace the tedious relocate-and-recollect data collection procedure with automatic demonstration generation. These methods only require a few human-collected data, based on which they augment object configurations

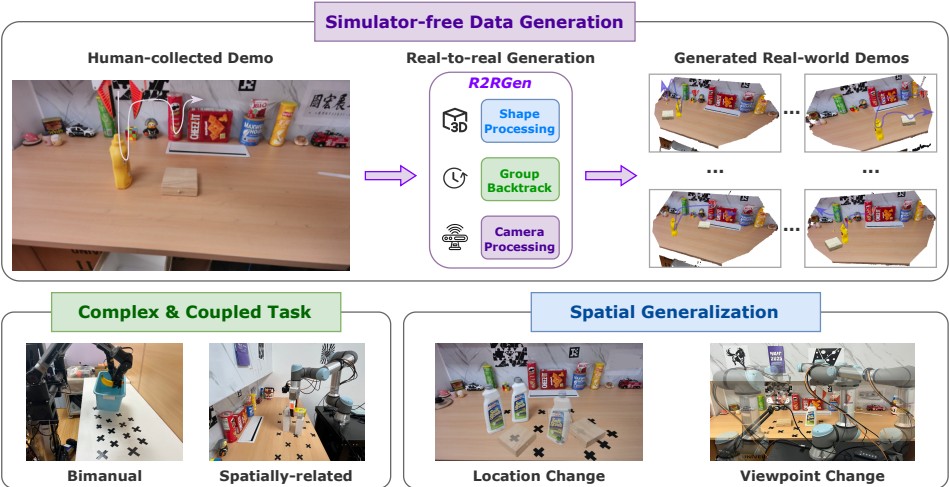

Figure 1: R2RGen is a simulator-free data generation framework. Given one human-collected demonstration, R2RGen directly parses and edits both pointcloud observations and action trajectories in a shared 3D space. R2RGen achieves strong spatial generalization on diverse complex tasks.

and apply transformation and interpolation to generate diverse trajectories with different motion patterns. Though achieving satisfactory performance in sumulation, these methods require on-robot rollouts to collect real-world observation-action pair, which takes much more time and relies on human supervision. Recently, DemoGen Xue et al. (2025) introduces a 3D-based data generation method that builds on point-cloud input policy Ze et al. (2024b). By operating directly in the 3D domain, the approach augments object point clouds to synthesize varied trajectories along with their corresponding visual observations. This pipeline is simulator- and rendering-free, thus being very efficient and avoiding sim-to-real gap. However, DemoGen exhibits several critical limitations that restrict its practical use: (1) It focuses on fixed-base setting, so viewpoint change is not taken into consideration; (2) it imposes strong assumption on input data, where the pointcloud of environment should be cropped, up to 2 objects are supported, and each skill must involve only one target object; (3) it suffers from visual mismatch problem, i.e., large augmentation leads to incomplete pointcloud observation. Due to these constraints, DemoGen does not fully achieve practical real-to-real generation and remains limited in handling mobile manipulation and diverse task configurations.

In this paper, we propose R2RGen, a real-to-real 3D data generation framework which is applicable for mobile robot Fu et al. (2024), works on raw pointcloud observation and supports any number of objects and interaction modes. Given a pre-segmented source demonstration, previous methods apply spatial transformation on each object individually. This object-centric paradigm can only handle skills relevant to only one target object. To overcome this problem, we propose group-wise data augmentation, which links each skill to a group of objects rather than a single target to maintain necessary object combination for complicated skills. It also leverages a backtracking mechanism to augment the 3D observation without disturbing the causal order of each operation. Moreover, since pointcloud from 3D sensor (e.g. RGB-D camera) is incomplete, large transformation (especially rotation) will make the augmented 3D observation unreasonble. E.g., points should be observed are missing, while points should be occluded exists. To this end, we further present camera-aware 3D post-processing to ensure the 3D observation after augmentation obey the distribution of 3D sensor. Through extensive real-world evaluation, we show that R2RGen, trained with only one human demonstration, outperforms policies trained with $25\times$ more human-collected data. Furthermore, R2RGen exhibits strong scalability with additional demonstrations, generalizes beyond spatial variation, and demonstrates promising potential for applications such as mobile manipulation.

## 2 RELATED WORK

**Imitation learning for robotic manipulation:** With the development of robotic data collection system and model architecture, using imitation learning to train visuomotor policies Zhao et al.

(2023); Chi et al. (2023); Prasad et al. (2024); Wang et al. (2024a;b); Ze et al. (2024b;a), which end-to-end predict actions from visual observation, becomes a promising way to learn dexterous manipulation skills from human demonstrations. Inspired by the success of large language models (LLM) and vision language models (VLM), there are multiple recent works exploring scaling up the model size and data amount to train generalist robot policies with imitation learning. One promising approach for training such generalist are vision-language-action models (VLA) Brohan et al. (2023); Kim et al. (2024b); Black et al. (2024); Cheang et al. (2024); Zhang et al. (2024); Cheng et al. (2024a); Li et al. (2024), which finetunes VLM pre-trained on internet-scale data for robot control. Though being flexible, imitation learning methods are data-intensive due to the lack of skill priors. In fact, to achieve strong generalization ability, visuomotor policy requires large amount of data for training / finetuning. Since largest embodied datasets O'Neill et al. (2024); Khazatsky et al. (2024) are still much smaller than the counterparts in vision and language fields Deng et al. (2009); Schuhmann et al. (2022), current works manage to solve this problem with advanced data collection systems like UMI Chi et al. (2024); Ha et al. (2024) and VR Cheng et al. (2024b); Ding et al. (2024), or empirical studies on data scaling Zhao et al. (2024); Lin et al. (2024); Zha et al. (2025) and data selection Hejna et al. (2024); Zhang et al. (2025) techniques.

**Data generation for visuomotor policy:** In order to train generalized manipulation visuomotor policy with less human labor, automatic data generation has been paid increased attention in recent years. A branch of works Hua et al. (2024); Wang et al. (2023a;b); Katara et al. (2024) utilize the common knowledge from LLM / VLM and privileged information from simulatorfor zero-shot task and motion planning. To improve data quality, some works generate robotic data from human demonstration video Duan et al. (2023); Lepert et al. (2025); Yu et al. (2025), which fully exploits structured skill primitives from human to acquire reasonable data. However, these methods still rely on vision foundation models Kim et al. (2024a); Kerr et al. (2024) to estimate and track poses of hand and object / part, which may not be accurate enough. Different from generating robotic data in robot-free manner, MimicGen Mandlekar et al. (2023) and its follow-up works Hoque et al. (2024); Garrett et al. (2024); Jiang et al. (2024) expand real-world demonstrations acquired from teleoperation by synthesizing different execution plans in simulator. These methods work well for simulation, but suffer from time-consuming on-robot rollouts to acquire real-world observation-action pairs. More recently, DemoGen Xue et al. (2025) proposes to apply augmentation on real-world 3D visual input as well as the trajectory. By using 3D policy, DemoGen directly takes the augmented 3D pointcloud as input, thus being simulator- and rendering-free. The real-to-real generation paradigm is efficient and plug-and-play without simulator setup, but currently DemoGen struggles on strong input assumption and visual mismatch problems which severely hinder its application.

## 3 APPROACH

### 3.1 PROBLEM STATEMENT

**Visuomotor policy learning:** Robotic manipulation task can be modeled as a Partially Observable Markow Decision Process (POMDP) with visuomotor policy $\pi : \mathcal{O} \mapsto \mathcal{A}$, which defines a function that maps current RGB-D observation $o_t \in \mathcal{O}$ to the robot's action $a_t \in \mathcal{A}$. $o_t = (I_t, P_t)$, where $I_t$ is RGB observation and $P_t$ is the pointcloud in camera coordinate system lifted from depth observation and camera intrinsics. To train this policy, a large dataset of demonstrations $\mathcal{D} = \{o_1^i, a_1^i, ..., o_{H_i}^i, a_{H_i}^i\}_{i=1}^N$ should be collected. To reduce human labor, we aim to improve data efficiency by generating spatially diverse data $\mathcal{D}'$ with only one human-collected source demonstration $D_s \in \mathcal{D}$. Formally written as:

$$\mathcal{D}' = \{D_s, D_g^1, D_g^2, ..., D_g^N\}, \ \ \{D_g^i\}_{i=1}^N = \text{R2RGen}(D_s) \tag{1}$$

It is expected that we can train a spatially generalized visuomotor policy purely from $\mathcal{D}'$. As a real-to-real framework, we directly augment the 3D observation as well as the action trajectory to generate diverse observation-action pair. Therefore, $\pi$ should be a 3D policy that directly takes in pointcloud as visual input. We opt for iDP3 Ze et al. (2024a) as our policy, which consumes the egocentric pointcloud $P_t$ without requirement on camera pose.

**Assumptions:** We make the following assumptions: (1) The visuomotor policy only predicts the actions of robotic arm. Although we support manipulation with different base positions, the mobile base remains fixed during each individual task execution. (2) The visual observation is captured with

a RGB-D camera, which should be fixed along with the base during task execution. (3) Similar to Mandlekar et al. (2023), the actions in a demonstration can be treated as a sequence of continuous end effector pose and discrete gripper state. Formally, $a_t = (\mathbf{A}_t^{ee}, a_t^{grip})$, where $\mathbf{A}_t^{ee}$ is SE(3) end-effector pose. We align the coordinate system of $\mathbf{A}_t^{ee}$ with the camera coordinate system to maintain a consistent representation.

### 3.2 SOURCE DEMONSTRATION PRE-PROCESSING

Given the source demonstration $D_s = \{o_1, a_1, ..., o_{H_s}, a_{H_s}\}$, we need to fully parse the 3D observations $\{P_t\}$ into editable composition of objects, and parse the action trajectory $\{a_t\}$ into motion and skill segments, which facilitates further data generation. Formally, the goal of this pre-processing stage is: (1) **Scene parsing**. Assume the task of $D_s$ involves $K$ relevant objects. So for each timestamp $t$, the pointcloud observation $o_t$ should be segmented into $K$ object pointclouds $\{P_t^1, ..., P_t^K\}$, one environment pointcloud $P_t^e$ and the pointcloud $P_t^a$ of robot's arm. (2) **Trajectory parsing**. Same with previous definition Garrett et al. (2024); Xue et al. (2025) of motion and skill, $\{a_t\}$ should be segmented into sequence of interleaved motion / skill segments $\{a_1, ..., a_{m_1}\}$ (motion-1), $\{a_{m_1+1}, ..., a_{s_1}\}$ (skill-1), $\{a_{s_1+1}, ..., a_{m_2}\}$ (motion-2), $\{a_{m_2+1}, ..., a_{s_2}\}$ (skill-2), etc. The objects relevant to each skill should also be known.

For scene parsing, we can segment the $K$ objects in the first frame $I_1$ and track through the whole video. The 2D masks can be projected into 3D to obtain pointcloud of each object. However, only segmenting each objects from $\{P_t\}$ is insufficient due to the incompleteness of RGB-D observation. For instance, the pointcloud of a cup may only represent the side facing the camera, leaving the occluded side unobserved. As a result, the observation will be incomplete when generating demonstrations from a novel viewpoint relative to the object. To this end, in addition to segmentation, we further complete the object pointclouds $\widetilde{P}_t^i = \mathcal{C}(P_t^i)$. We adopt a template-based 3D object tracking system Wen et al. (2024) to achieve this, which generates complete object point-

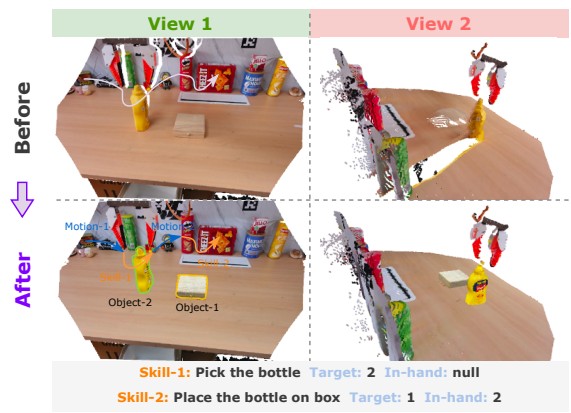

Figure 2: Pre-processing results. The 3D scene is parsed into complete objects, environment and robot's arm. The trajectory is parsed into interleaved motion and skill segments.

clouds $\{\widetilde{P}_t^1, ..., \widetilde{P}_t^K\}_{t=1}^{H_s}$ for all frames given $K$ object masks in the first frame $I_1$. We also need to acquire the complete environment pointcloud $\widetilde{P}^e$. Since the camera is fixed during task execution, $\widetilde{P}^e$ is static regardless of time. Therefore, we can remove the $K$ objects and get an observation $o_0$ before we collect $D_s$. Then simply set $\widetilde{P}^e$ as $P_0$. Given the complete objects $\{\widetilde{P}_t^1, ..., \widetilde{P}_t^K\}$ and environment $\widetilde{P}^e$, the arm pointcloud $P_t^a$ can be obtained by $P_t^a = P_t \setminus (\widetilde{P}^e \cup \widetilde{P}_t^1 \cup ... \cup \widetilde{P}_t^K)$. We do not complete $P_t^a$ since we empirically find it brings negligible influence. As a result, the pointcloud observation $P_t$ for each frame is parsed into the combination of complete objects, environment and robot's arm.

For trajectory parsing, we introduce a lightweight annotation system. The interface plays the RGB video $\{I_1, ..., I_{H_s}\}$ and asks the annotator to label the start frame and end frame of each skill. The intermediate trajectories between two skill segments are classified as motion segments. Apart from annotating skill segments, the annotator also specifies the object IDs (ranging from 1 to $K$) associated with each skill. Specifically, for every skill, both target object IDs and in-hand object ID are provided: the in-hand object refers to the item being held by the gripper during the skill execution (if any), while the target objects denote the entity with which the gripper interacts. IDs may be null, a single object, or multiple objects, depending on the task structure. The entire process uses only the RGB video as input and requires less than 60 seconds per demonstration, making it efficient and minimally labor-intensive—particularly suitable for settings with very few source demonstrations.

Figure 2 illustrates the parsed results of a source demonstration. More details about the object parsing and trajectory parsing systems can be found in Appendix A.1.

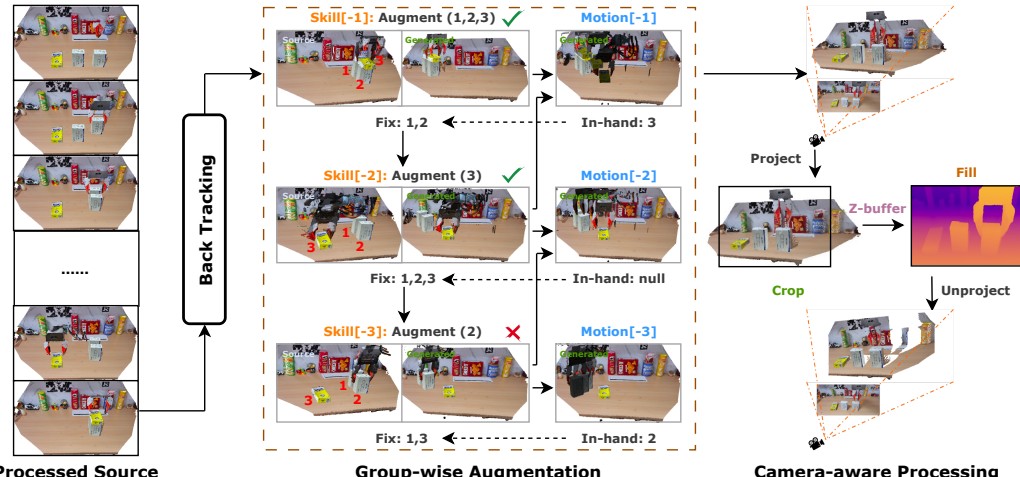

Figure 3: The pipeline of *R2RGen*. Given processed source demonstration, we backtrack skills and apply group-wise augmentation to maintain the spatial relationships among target objects, where a fixed object set is maintained to judge whether the augmentation is applicable. Then motion planning is performed to generate trajectories that connect adjacent skills. After augmentation, we perform camera-aware processing to make the pointclouds follow distribution of RGB-D camera. The solid arrows indicate the processing flow, while the dashed arrows indicate the updating of fixed object set.

### 3.3 GROUP-WISE DATA AUGMENTATION

To synthesize new demonstrations, we randomly augment the location and rotation of objects and environment to acquire new scene configurations, and generate the corresponding action trajectories. Previous pointcloud-based data generation methods Xue et al. (2025) assume only one target object per skill and transform skill segments solely based on that object's transformation. Motion segments are then generated using a planner to connect adjacent skills into a complete trajectory. However, this approach fails when a skill involves multiple target objects whose spatial relationships must be preserved. For instance, for task "build a bridge" shown in Figure 3, placing the bridge deck (object-3) requires the two bridge piers (object-1 and object-2) to be positioned at a specific relative distance. Independently augmenting each pier would disrupt this relationship and prevent successful execution of the final skill. To support arbitrary interaction modes, we propose a group-wise augmentation strategy that maintains structural constraints among multiple objects during data generation.

**Group-wise backtracking:** Instead of modeling skills as object-centric, we assign each skill to a group of objects consists of the annotated target and in-hand objects. All objects within the same group undergo identical geometric transformations (i.e., the same translation and rotation). Augmentations are performed in a backtracking manner to avoid causal conflicts among object states. Formally, we begin from the last skill (skill-$n$) $\{a_{m_n+1}, ..., a_{s_n}\}$. Denote $O_n = O_n^{tar} \cup O_n^{hand}$ is the ID set of target and in-hand objects of current skill, and $\overline{O}_n = \varnothing$ is the ID set of fixed objects (i.e., objects cannot be augmented). We decide whether to augment current group according to $\overline{O}_n \cap O_n$. If the intersection is $\varnothing$, we randomly sample a transformation matrix $\mathbf{T}_n \in \mathbb{R}^{4 \times 4}$ to apply XY-plane translation and Z-axis rotation on group $O_n$ (XY plane is fitted through the tabletop point cloud). Otherwise this group is fixed at current time and we cannot augment it. After applying the group transformation, the fixed object set is updated as follows:

$$\overline{O}_{n-1} = (\overline{O}_n \cup O_n^{tar}) \setminus O_n^{hand} \qquad (2)$$

where current group is appended into the fixed set to maintain spatial relationships, while in-hand object is released since its state before grasped is independent of current skill's constraints. We then proceed to skill-$(n-1)$ and repeat above operations until all skills are traversed.

**Skill augmentation:** For skill-$i$, if the corresponding group is not fixed, we apply transformation $\mathbf{T}_i$ to augment the end effector's pose while remain the gripper state:

$$\hat{\mathbf{A}}_t^{ee} = \mathbf{A}_t^{ee} \cdot \mathbf{T}_i, \ \ \hat{a}_t^{grip} = a_t^{grip}, \ \ \forall t \in [m_i + 1, s_i] \qquad (3)$$

Then the pointclouds of objects $\widetilde{P}_t^k$ ($k \in O_i$) and arm $P_t^a$ are transformed with $\mathbf{T}_i$ in the same way.

**Motion augmentation:** For motion-$i$, we apply motion planning to generate a trajectory that starts at the end of skill-$(i-1)$ and ends at the beginning of skill-$i$:

$$\hat{\mathbf{A}}_{t_1:t_2}^{ee} = \texttt{MotionPlan}(\hat{\mathbf{A}}_{s_{i-1}}, \hat{\mathbf{A}}_{m_i+1}), \ \ \hat{a}_{t_1:t_2}^{grip} = a_{t_1:t_2}^{grip}, \ \ t_1 = s_{i-1} + 1, \ \ t_2 = m_i \quad (4)$$

Then the pointclouds of in-hand objects $\widetilde{P}_t^k$ ($k \in O_i^{hand}$) and arm $P_t^a$ are transformed with the relative pose transformation $(\mathbf{A}_t^{ee})^{-1} \cdot \hat{\mathbf{A}}_t^{ee}$.

Finally, we apply a random transformation $\mathbf{T}_e$ on the environment $\widetilde{P}^e$ for all frames, which simulates the viewpoint change of robot. An algorithm diagram of the group-wise augmentation pipeline is detailed in Appendix A.2. For bimanual manipulation, we additionally introduce constraints to ensure the generated demonstrations executable. Refer to Appendix A.3 for more details on bimanual tasks.

### 3.4 CAMERA-AWARE 3D POST-PROCESSING

After training on generated demonstrations $\mathcal{D}'$, the 3D policy is deployed in real-world with RGB-D camera as input sensor. Therefore, the pointcloud observation in $\mathcal{D}'$ should be similar to the raw RGB-D observation. Currently, there are two main differences: (1) The generated pointcloud is over-complete. While for a given perspective of RGB-D camera, raw pointcloud converted from depth image is complete only at this viewpoint. (2) Due to the augmentation on environment, the spatial distribution of generated pointcloud is shifted. To solve this, we propose a camera-aware 3D post-processing to adjust the distribution of generated pointcloud observations $\{\hat{P}_t\}$:

$$\hat{P}_t^{adjust} = \mathcal{P}^{-1}(\texttt{Fill}(\texttt{Z-buffer}(\texttt{Crop}(\{(u_i, v_i, d_i)\})))), \ \ \{(u_i, v_i, d_i)\} = \mathcal{P}(\hat{P}_t) \quad (5)$$

where $\mathcal{P}$ projects 3D pointcloud to image plane with camera intrinsics. The $\texttt{Crop}$ operation removes pixels $\{(u_i, v_i, d_i) | u_i < 0 \ or \ u_i \geq W \ or \ v_i < 0 \ or \ v_i \geq H\}$ which are out of image boundary. $\texttt{Z-buffer}$ processes overlapped pixels and only keep one pixel with smallest depth value, which removes hidden points at current viewpoint. In practice, we notice the density of pointclouds may not be so high, making front surface unable to hide all points behind. Therefore, we propose a patch-wise Z-buffer operator, where each point with small depth value can hide deeper points in a $r$-radius neighborhood on image plane. Since the environment is augmented, pixels near the image boundary may be empty (i.e., no point is projected to these pixels). So we $\texttt{Fill}$ the empty pixels by either shrinking the image size or expanding the environment pointcloud, which we detail in Section 4.4. Finally, after post-processing in the image plane, we project the pixels back to camera coordinate system. The adjusted pointcloud $\hat{P}_t^{adjust}$ well matches the distribution of RGB-D camera and can be directly fed into our 3D policy during training.

## 4 EXPERIMENT

In this section, we evaluate R2RGen through extensive real-world experiments, demonstrating its effectiveness on one-shot imitation learning and how it scales up with more source demonstrations. We also conduct comprehensive ablation studies to explore the optimal design choices. Furthermore, we show R2RGen can be extended to facilitate appearance generalization and mobile manipulation.

### 4.1 EXPERIMENTAL SETUP

**Policy.** We select iDP3 Ze et al. (2024a) as the visuomotor policy, which takes egocentric pointclouds (i.e., pointclouds in camera coordinate system) and proprioception state as inputs without requirement on camera pose and calibration. Details on training iDP3 on each task are described in Appendix A.4.

**Hardware.** We utlize two robot platforms: single-arm and bimanual. The single-arm platform consists of a 7-DoF UR5 arm equipped with a parallel jaw gripper and a mobile base. An ORBBEC femto bolt camera is rigidly affixed via a mounting bracket to the mobile base, which provides the RGB-D observations. The bimanual platform adheres to the MobileAloha architecture Fu et al. (2024), employing dual AgileX PiPER arms integrated with HexFellow omnidirectional mobile base. A head-mounted RGB-D camera provides egocentric perception. See Appendix B for further details.

Table 1: Real-world evaluation of *R2RGen* for spatial generalization. Success rate is reported.

| | Single-Arm Task | | | | | | Dual-Arm Task | | Averaged |
| | Open-Jar | Place-Bottle | Pot-Food | Hang-Cup | Stack-Brick | Build-Bridge | Grasp-Box | Store-Item | |
|---|---|---|---|---|---|---|---|---|---|
| 1 Source | 3.1 | 3.1 | 3.1 | 3.1 | 3.1 | 3.1 | 4.2 | 4.2 | 3.4 |
| +DemoGen | 18.8 | 15.6 | – | – | – | – | 16.7 | 16.7 | – |
| **+R2RGen** | **50.0** | **50.0** | **37.5** | **34.4** | **43.8** | **34.4** | **41.7** | **33.3** | **40.3** |
| 10 Source | 56.3 | 34.3 | 9.4 | 15.6 | 9.4 | 9.4 | 25.0 | 20.8 | 22.5 |
| 25 Source | 78.1 | 53.1 | 21.9 | 43.8 | 40.6 | 28.1 | 29.2 | 33.3 | 41.0 |
| 40 Source | 87.5 | 68.8 | 28.1 | 43.8 | 50.0 | 43.8 | 37.5 | 41.7 | 50.2 |

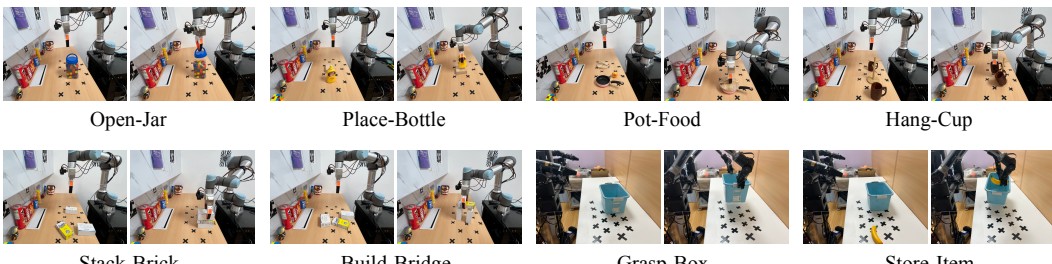

| Open-Jar | Place-Bottle | Pot-Food | Hang-Cup |
| Stack-Brick | Build-Bridge | Grasp-Box | Store-Item |

Figure 4: Visualization of our real-world tasks. We show the start and end moments of each task.

**Tasks and evaluation.** We design 8 representative tasks for evaluation, including 2 simple tasks (Open-Jar, Place-Bottle), 4 complex tasks (Pot-Food, Hang-Cup, Stack-Brick, Build-Bridge) and 2 bimanual tasks (Grasp-Box, Store-Item), as visualized in Figure 4. We compare with DemoGen Xue et al. (2025) on the two simple single-arm tasks and two bimanual tasks. While for other tasks involving complex spatial relationships among objects, DemoGen fails to generate reasonable data. We evaluate different methods on diverse objects' locations and rotations as well as robot's viewpoints, including a portion of out-of-distribution samples not encountered during training. Refer to Appendix C for detailed task definition and evaluation protocol.

## 4.2 RESULTS: ONE-SHOT IMITATION LEARNING

For one-shot imitation learning, we only collect one human demonstration for R2RGen to generate new data. Similar to DemoGen, we replay the collected human demonstration twice to acquire diverse pointcloud observations, which significantly reduce the impact of sensor noise. The three pointcloud trajectories are all used for 3D data generation. Then we compare the policy trained on purely generated data with ones trained on different number of human demonstrations, as shown in Table 1. When trained with only one human demonstration, the policy succeeds merely at the demonstrated pose but fails to generalize. Both DemoGen and R2RGen improve its performance. It is shown that R2RGen consistently outperforms DemoGen across all tasks, even though DemoGen crops pointclouds of background while we do not. This advantage primarily stems from our scene parsing and camera-aware processing techniques, which enable generating high-quality data under large variations in object location / rotation and robot viewpoint. In contrast, DemoGen suffers from significant visual mismatch under such challenging evaluation. R2RGen achieves performance comparable to policies trained with 25 human demonstrations, and even surpasses 40 demonstrations on several difficult tasks, which validates its effectiveness on spatial generalization.

## 4.3 RESULTS: PERFORMANCE-ANNOTATION TRADEOFF

We further study how R2RGen scales up with more source demonstrations. For each task, we run R2RGen to generate data from 1, 2, 3 and 5 human demonstrations respectively. We then report the policy performance boost w.r.t. the increase of synthetic data under different numbers of source demonstrations. As shown in Figure 5, the success rate gradually saturates as the number of generated demonstrations increases—a trend also observed with human-collected data. This behavior is due to the limited capacity of the iDP3 policy, which uses a lightweight PointNet encoder. Further scaling beyond this plateau would require policies with larger 3D backbones. We also note that more source demonstrations lead to a higher saturation performance, demonstrating R2RGen's ability to effectively leverage additional data for improved results.

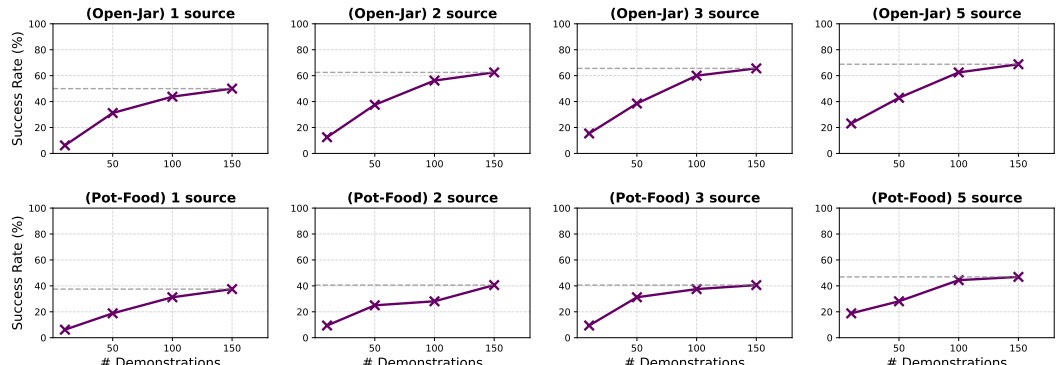

Figure 5: Effects of the number of generated demonstrations and source demonstrations on the final performance of R2RGen.

Table 2: Effects of the pointcloud processing.

| Method | SR |
|---|---|
| Remove pointcloud completion | 12.5 |
| Remove environment pointcloud | 18.8 |
| Remove environment augmentation | 28.1 |
| *R2RGen* | **50.0** |

Table 3: Effects of camera-aware processing.

| Method | SR |
|---|---|
| Remove `Crop` operation | 34.4 |
| Remove `Z-buffer` operation | 15.6 |
| Remove `Fill` operation | 28.1 |
| *R2RGen* | **50.0** |

## 4.4 ABLATION STUDY

We further conduct ablation experiments to study the effects of each design choice. The analysis is conducted on the Place-Bottle task.

**Pointcloud processing.** We first study how object and environment pointclouds affect the final performance, as shown in Table 2. We notice removing pointcloud completion will lead to unrealistic data under large spatial augmentation, while removing operations on environment reduces the policy's robustness to viewpoint changes.

**Camera-aware processing.** We then ablate camera-aware processing by removing one of the key operations at each time. As shown in Table 3, these operations play a critical role in determining final performance. For the `Fill` operation, we further compare

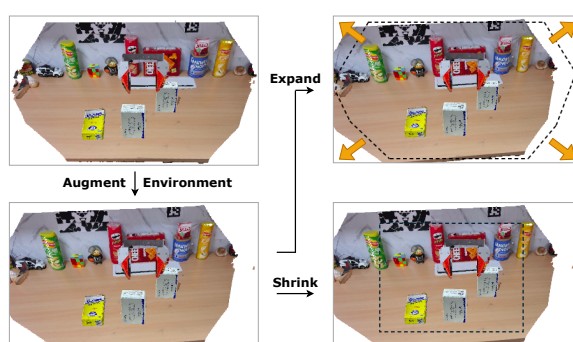

Figure 6: Two implementations of `Fill` operation, i.e., shrinking and expanding.

two design choices in Figure 6: by shrinking the image size or expanding the environment pointcloud to fill the black dashed outline (i.e., the valid-depth area of the RGB-D camera). If shrinking is adopted, we will apply the same shrinking to raw RGB-D observations when deploying the policy trained with R2RGen. Empirically, both methods achieve comparable performance; we ultimately adopt shrinking due to its operational simplicity and lack of additional processing requirements.

## 4.5 EXTENSION AND APPLICATION

**Extension: appearance generalization.** Beyond spatial generalization, robotic manipulation tasks involve other forms of generalization, such as appearance generalization (i.e., adapting to novel object instances and environments) and task generalization. Among these, spatial generalization serves as the fundamental prerequisite for other generalization capabilities. Since this work focuses on single-task visuomotor policy learning, we investigate whether the spatial generalization enabled by R2RGen can further facilitate appearance generalization. As shown in Figure 7, we design a more

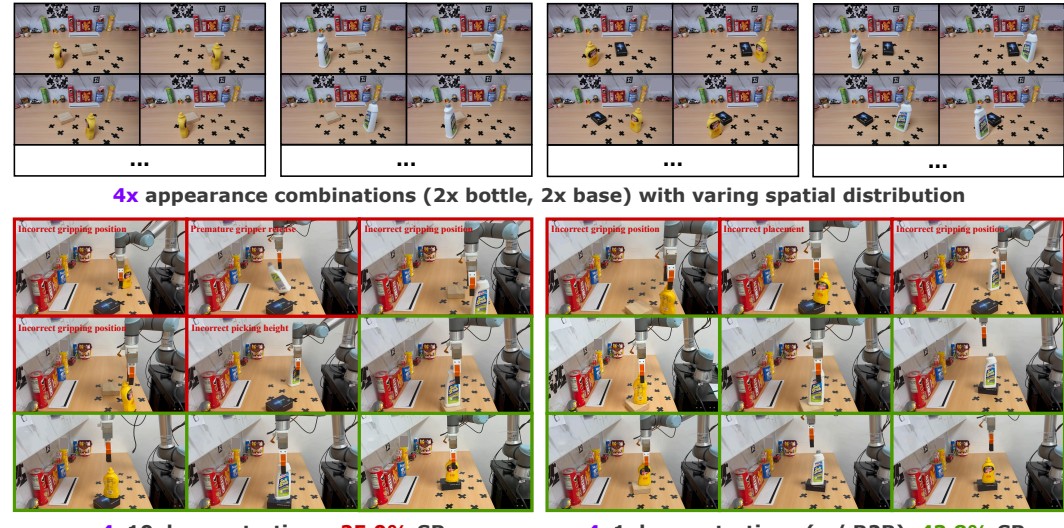

**4x** appearance combinations (2x bottle, 2x base) with varing spatial distribution

**4x10** demonstrations: **25.0%** SR      **4x1** demonstrations (w/ R2R): **43.8%** SR

Figure 7: Extension on appearance generalization. The spatial generalization of R2RGen can serve as a foundation to achieve other kinds of generalization with much less data.

challenging Place-Bottle task with four distinct bottle-base appearance combinations (2 bottle types × 2 base types). We observe that achieving both appearance and spatial generalization significantly increases data demand. Even with 40 human demonstrations (10 per bottle-base pair), the policy only reaches a 25% success rate. In contrast, using R2RGen, only 1 demonstration per bottle-base pair (4 in total) is needed to achieve a success rate of 43.8%, demonstrating its efficiency in handling combined generalization challenges.

**Application: mobile manipulation.** The strong spatial generalization ability achieved by R2RGen can also facilitate the challenging mobile manipulation task. Being generalized to different camera views, we solve mobile manipulation as a simple combination of navigation and fixed-base manipulation. Here we adopt MoTo Wu et al. (2025) as the navigation strategy to approach the target object, and then apply a policy trained with R2RGen to complete the task. The experimental video demo is shown in our project page. Refer to Appendix D for more details.

## 5 CONCLUDING REMARK

R2RGen introduces a real-to-real 3D data generation framework that generalizes beyond prior pointcloud-based methods such as DemoGen Xue et al. (2025). Specifically, it supports mobile manipulators, raw sensor inputs, arbitrary numbers of objects, and diverse interaction modes, overcoming key limitations of existing approaches. With only one human demonstration, our method directly augments the 3D observation as well as the action trajectories to generate large amount of pointcloud-action pairs, which are utilized to train a spatially generalized 3D policy. The overall pipeline is carefully designed with user-friendly pre-processing, group-wise augmentation and camera-aware post-processing, which ensures the visual observations are complete and spatial relationships between objects are maintained. Extensive experiments on multiple real-world tasks validate the effectiveness of R2RGen. We further extend its application to appearance generalization and mobile manipulation scenarios, demonstrating its strong generalizability and potential for broad real-world deployment.

**Potential Limitations.** Currently, there are two major limitations of R2RGen. (1) The RGB-D sensor should be fixed during task execution. So R2RGen cannot be applied to source demonstration collected with wrist camera or moving base. (2) During pre-processing of source demonstration, we apply a template-based 3D object tracking model Wen et al. (2024) to acquire complete object pointclouds for all frames, which only works for rigid objects. For non-rigid objects, please refer to Appendix A.1 for more details.

## REPRODUCIBILITY STATEMENT

To support reproducibility, we provide a detailed description of the R2RGen framework in the main text and appendix, including algorithm outline and implementation specifics. A minimal runnable example demonstrating the core data generation process is available on our anonymous project page: https://r2rgen.github.io. This demo allows users to execute key stages of R2RGen in a predefined scenario and adjust hyperparameters interactively. Due to the integrated nature of the full pipeline—which involves template reconstruction, data annotation, trajectory generation, and real-robot deployment—the complete code is still being cleaned. We commit to releasing the full codebase (covering both R2RGen method and real-robot deployment) upon acceptance of the paper.

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

APPENDIX

Please visit our anonymous website to view robot videos: https://r2rgen.github.io.

# A IMPLEMENTATION DETAILS

## A.1 ANNOTATION SYSTEM

There are three stages to collect and process source demonstration, as shown below.

**Template and environment scanning:** To collect demonstration, we first move the robot to the front of a table or any other platform. Then the robot's base and head camera are fixed during task execution. Before we perform teleoperation, we first remove all relevant objects from the table and take a RGB-D image $o_0$ as the complete environment. Then we individually scan each object to acquire their 3D template via the RealityComposer App on iPad.

**Annotation with UI:** After scanning, we put objects back to the table and start teleoperation to collect sequence of observation-action pair. After that, our UI plays the RGB video and ask user to annotate. As shown in Figure 8. The user first draws boxes on the initial frame to label each object with index, which is then processed with SAM Kirillov et al. (2023) to get object masks. Then user watches the video and is able to click `Play`, `Stop` or `Rollback` at anytime to capture key frames (i.e., the start / end of skill segments). When the user stops at a key frame, they can press `Annotate` and enter annotation mode. In this mode, the user is asked to type in the start and end frame of each skill segment, as well as the target and in-hand object IDs corresponding to each skill. The annotations will be processed into a json file. We show an example as below.

```json
{
  "masks": [
    "mask_gripper.png",
    "mask_1.png",
    "mask_2.png"
    "mask_3.png"
  ],
  "arms": 2,
  "annotations": [
    {
      "frame": 4,
      "type": "motion"
    },
    {
      "frame": 12,
      "type": "skill",
      "target": [2],
      "left_hand": null,
      "right_hand": null
    }
    {
      "frame": 23,
      "type": "motion"
    },
    {
      "frame": 31,
      "type": "skill",
      "target": [1,3],
      "left_hand": [2],
      "right_hand": null
    }
  ]
}
```

**Object tracking and completion:** With the object masks in first frame, the whole RGB-D video and the 3D templates of all objects, we run FoundationPose Wen et al. (2024) to track each object across all frames. FoundationPose can accurately predict the 6-DoF pose of each object. So we use the pose to transform object template into world coordinate system to acquire complete object pointclouds.

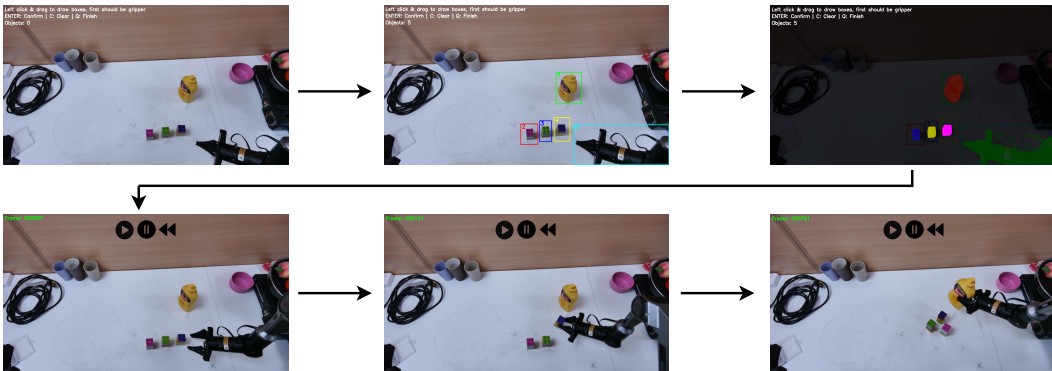

Figure 8: The annotation UI. The users first segment all relevant objects in the first frame. Then they click `Play`, `Stop` or `Rollback` to capture key frames for skill / motion annotation.

With the pointclouds of complete enironments $\widetilde{P}^e$ and complete objects $\{\{\widetilde{P}_t^1, ..., \widetilde{P}_t^K\}\}$, we finally calculate set difference between the above complete pointclouds and the raw observation to isolate the robot's arm $\{P_t^a\}$ for all frames.

**Parsing non-rigid objects:** Our R2RGen is a general framework that can also handle non-rigid objects. However, since currently FoundationPose only supports rigid objects, for non-rigid ones (assume with ID $J$), we instead apply SAM2 Ravi et al. (2024) to track and segment the object across all frames. Then we query the object pointclouds $\{P_t^J\}$ from the RGB-D images according to the object masks. With this simple modification, our framework is able to support non-rigid objects as well. Different from $\{\{\widetilde{P}_t^1, ..., \widetilde{P}_t^K\}\}$, $\{P_t^J\}$ are incomplete pointclouds, so strong augmentation on them will lead to visual mismatch as in DemoGen. Therefore, we only apply weak augmentation on non-rigid objects and skip camera-aware processing on their pointclouds. This limitation may be solved by 3D pointcloud completion methods or an improved FoundationPose model, which we leave for future work.

### A.2 PIPELINE OF AUGMENTATION

An algorithm diagram of the group-wise augmentation pipeline is shown in Algorithm 1.

To augment pointcloud observations, we can divide the pointcloud of whole scene into three parts, i.e., environments, robot's arm and objects:

$$\hat{P}_t = \widetilde{P}^e \cdot \mathbf{T}_e \ \cup \ P_t^a \cdot (\mathbf{A}_t^{ee})^{-1} \cdot \hat{\mathbf{A}}_t^{ee} \ \cup \ \texttt{ObjectAugment}_{k=1}^K(\widetilde{P}_t^k, \{\hat{a}_t\}) \tag{6}$$

Here the environment pointcloud is static across all frames. The arm pointcloud is correlated to the pose of end effector. However, different objects are involved in different motion and skill segments, which cannot be augmented as a whole. Therefore, we traverse all objects. For object $k$, we backtrack all skills that contain this object: $\mathcal{S}_{-1}^k, \mathcal{S}_{-2}^k, ...$ Then we augment $\widetilde{P}_t^k$ according to the timestamp:

$$\hat{P}_t^k = \begin{cases} \widetilde{P}_t^k \cdot (\mathbf{A}_t^{ee})^{-1} \cdot \hat{\mathbf{A}}_t^{ee} & , t \in \texttt{MotionHand}(k) \\ \widetilde{P}_t^k \cdot \mathbf{T}_{-1}^k & , \text{otherwise} \end{cases} , \ \text{End}(\mathcal{S}_{-2}^k) < t \tag{7}$$

$$\hat{P}_t^k = \begin{cases} \widetilde{P}_t^k \cdot (\mathbf{A}_t^{ee})^{-1} \cdot \hat{\mathbf{A}}_t^{ee} & , t \in \texttt{MotionHand}(k) \\ \widetilde{P}_t^k \cdot \mathbf{T}_{-1}^k \cdot \mathbf{T}_{-2}^k & , \text{otherwise} \end{cases} , \ \text{End}(\mathcal{S}_{-3}^k) < t \leq \text{End}(\mathcal{S}_{-2}^k)$$

$$...$$

where $\text{End}(\cdot)$ means the end frame of a segment. If the segment does not exist, the value will be set to $-1$. $\texttt{MotionHand}(k)$ represents the set of timestamps that $k$ is in-hand during motion (the in-hand object of motion $\mathcal{M}_i$ equals to that of skill $\mathcal{S}_i$). During backtracking, every skill involving the target object triggers cumulative application of its spatial transformation to itself and prior timestamps.

---

**Algorithm 1:** Pipeline of Group-wise Augmentation.

---

**Input:** Trajectory of motion $\{\mathcal{M}_1, ..., \mathcal{M}_H\}$ and skill $\{\mathcal{S}_1, ..., \mathcal{S}_H\}$; Pointclouds of enironments $\widetilde{P}^e$, objects $\{\{\widetilde{P}_t^1, ..., \widetilde{P}_t^K\}\}$ and robot's arm $\{P_t^a\}$; Set of target objects $\{O_i^{tar}\}$ and in-hand object $\{O_i^{hand}\}$ of each skill $\mathcal{S}_i$; Transformation on each skill $\{\mathbf{T}_1, ..., \mathbf{T}_H\}$.

**Output:** Augmented 3D observation $\{\hat{P}_t\}$ and action $\{\hat{a}_t\}$.

Initialize fixed object set $\overline{O}_H = \varnothing$, backtracking index $T = H$;
// Backtrack skills
**while** $T > 0$ **do**
    // If current group is not fixed, augment the trajectory
    **if** $(O_T^{tar} \cup O_T^{hand}) \cap \overline{O}_T = \varnothing$ **then**
        Augment $S_T$ with Eq (3);
    // Otherwise just copy the trajectory
    **else**
        Set $\{\hat{a}_t | t \in \mathtt{Timestamp}(S_T)\} = S_T$;
    Update $\overline{O}_T$ with Eq (2);
    $T = T - 1$;
// Interpolate motions
**while** $T < H$ **do**
    Interpolate motion $\{\hat{a}_t | t \in \mathtt{Timestamp}(\mathcal{M}_{T+1})\}$ with Eq (4);
    $T = T + 1$;
// Augment 3D observations according to new trajectories
Acquire $\{\hat{P}_t\}$ with Eq (6);

---

## A.3 BIMANUAL TASKS

As a general framework, R2RGen can also support bimanual manipulation with less modification. (1) For source pre-processing: we do not segment skill and motion for each arm separately. We just extend the in-hand information to in-left-hand and in-right-hand when annotate each skill. In this way, both single-arm and dual-arm operations can be unified. Then for the arm pointcloud $P_t^a$, we cluster it into two parts to get pointclouds of left arm $P_t^{la}$ and right arm $P_t^{ra}$. (2) For group-wise augmentation: if in skill $\mathcal{S}_i$, in-left-hand and in-right-hand objects are the same (not null), this means this object is also held by both arms during motion $\mathcal{M}_i$. Therefore, the trajectories of both arms during $\mathcal{M}_i$ should follow a fixed spatial relationship to ensure the object can be stably grasped in both hands. For other cases, we individually interpolate the trajectories for both arm during motion.

## A.4 DATA GENERATION AND TRAINING

Since real-world data always has random noise, one source demonstration may not be enough to cover the distribution of pointclouds. In our experiments, we still collect only one human demonstration, but replay the action trajectory for three times as did in DemoGen. Then we generate demonstrations based on all three source demonstrations. For each new demonstration generated from a source demonstration, we randomly add small perturbations on the augmented locations and rotations for three times. Specifically, we add random tranlsation within a circle of $1.5$cm radius and random rotation within $\pm 20°$. The total number of generated demonstrations is calculated as $3 \times N \times 3$, where $N$ is the number of combinations of all augmented locations and rotations.

To train iDP3, denote $T_o$ as the observation horizon, $T_p$ as the action prediction horizon, and $T_a$ as the action execution horizon, we set $T_o = 2$, $T_p = 16$, and $T_a = 8$. The visuomotor policy is run at $5$ Hz. Training was performed for 6,000 epochs on a single RTX 4090 GPU (batch size 64) using Adam (learning rate $1 \times 10^{-4}$, weight decay $1 \times 10^{-6}$). Validation performance plateaued after approximately 2,500 epochs, and we selected the checkpoint with the lowest validation loss.

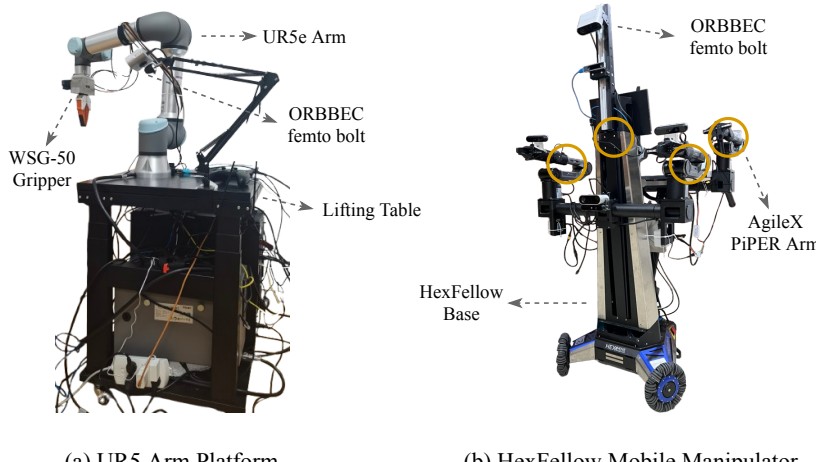

(a) UR5 Arm Platform     (b) HexFellow Mobile Manipulator

Figure 9: Robot platform overview. We employ two robot platforms: (a) single-arm UR5e system and (b) dual-arm Mobile Aloha system.

## B  HARDWARE SETUP

We utilize two robot platforms. The primary platform (Figure 9, a) is a single 7-DoF UR5e arm equipped with a Weiss WSG-50 parallel-jaw gripper. A ORBBEC femto bolt RGB-D camera is mounted beside to acquire visual observations. The arm is fixed on a height-adjustable table with movable base, which makes it possible for us to evaluate policy with different viewpoint and height. The action space is 7-dimensional (6-DoF end-effector pose plus gripper width). We use a Xbox controller to teleoperate the robotic arm to collect demonstrations.

The second platform (Figure 9, b) follows the design paradigm of Mobile Aloha Fu et al. (2024), using four AgileX PiPER Arms (two for teleoperation, two for manipulation) and a HexFellow omnidirectional mobile base. We mount one ORBBEC femto bolt RGB-D camera on the robot's head to acquire visual observation. Each robotic arm has 7 dimensions (6-DoF end-effector pose plus gripper width) and the overall action space is 14-dimensional.

## C  TASKS AND EVALUATIONS

### C.1  TASK DEFINITION

We carefully design 8 tasks for evaluation, including 6 single-arm tasks and 2 bimanual tasks. A task summary is provided in Table 4.

We describe these tasks in the text as follows, where skill and motion verbs are highlighted as orange and blue respectively:

**(A) Open-Jar.** The gripper moves above the jar and lowers to an appropriate height. Then it opens, moves down and closes to grasp the handle. It further rotates to open the jar.

**(B) Place-Bottle.** The gripper first moves to the bottle and grasps it. Then it lifts the bottle up and places it on the base.

Table 4: A summary of our real-world tasks. #Obj: number of manipulated objects. #Eval: number of evaluated configurations. #Demo: number of generated demonstrations.

| Task | Platform | #Obj | #Eval | #Demo |
|------|----------|------|-------|-------|
| **Open-Jar** | Single-arm | 1 | 32 | 144 |
| **Place-Bottle** | Single-arm | 2 | $4 \times 8$ | 144 |
| **Pot-Food** | Single-arm | 3 | $2 \times 4 \times 4$ | 144 |
| **Hang-Cup** | Single-arm | 3 | $2 \times 4 \times 4$ | 144 |
| **Stack-Brick** | Single-arm | 3 | $2 \times 4 \times 4$ | 144 |
| **Build-Bridge** | Single-arm | 3 | $2 \times 4 \times 4$ | 144 |
| **Grasp-Box** | Dual-arm | 1 | 24 | 108 |
| **Store-Item** | Dual-arm | 2 | $3 \times 8$ | 108 |

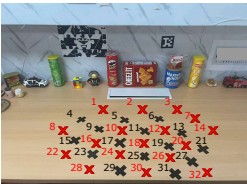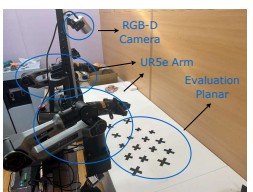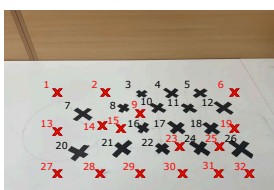

(a) Single-arm Evaluation  (b) Dual-arm Evaluation

Figure 10: Protocol for evaluating spatial generalization. We evaluate policies on different robot's viewpoints, object locations and rotations. Black crosses indicate seen locations (if human demonstrations are sufficient to cover) and red ones denote unseen locations during training.

**(C) Pot-Food.** The gripper first moves to the food and grasps it. Then it moves towards the pot and puts food into pot. Next it moves to the pot lid and picks it up. It finally moves towards the pot again and covers the pot with the lid.

**(D) Hang-Cup.** The gripper moves to the first cup and picks it up. Then it moves towards the shelf and hangs the cup on the shelf. It repeats the same operation on the second cup, but this time the cup should be hanged on a different position of the shelf.

**(E) Stack-Brick.** The gripper moves to the first brick and picks it up. Then the gripper moves to a designated place and places the brick there. It repeats the same operation on the other two bricks to stack them one-by-one.

**(F) Build-Bridge.** The gripper first moves to a white box and picks it up. Then it brings the box to a designated place and places it there. The same operation is repeated on the second white box, where the two white boxes (i.e., the bridge piers) should be placed in proper distance. Then the gripper moves to and grasps the yellow box (i.e., the bridge deck), moves towards the bridge piers and puts deck on piers to build a bridge.

**(G) Grasp-Box.** The left gripper moves to the left side of the box. Then the right gripper moves to the right side. After that, two grippers simultaneously grasp the box and lift it up.

**(H) Store-Item.** The left gripper moves to the left side of the box. Then it grasps the box and lifts it up. At the same time, the right gripper moves to the banana and grasps it. It then brings the banana to the box and stores the banana into box.

## C.2 EVALUATION PROTOCOL

To evaluate spatial generalization, we define large planar evaluation workspaces as illustrated in Figure 10. For each test trial, the initial positions of the objects are determined by sampling distinct locations from a pre-defined set of 32 points on the workspace. Each object is also assigned a random rotation sampled from the range of -20 to 20 degrees, while the robot's base is initialized at one of three distinct locations. For tasks involving more than one object, we constrain the range of locations for each object to reduce the overall number of combinations. We demonstrate the range of object locations of single-arm tasks in Figure 10 (a):

**(A) Open-Jar.** The range of jar is $\{1, 2, ..., 32\}$.

**(B) Place-Bottle.** The range of bottle is $\{4, 10, 8, 17, 22, 24, 28, 29\}$, and the range of box is $\{11, 13, 25, 26\}$.

**(C) Pot-Food.** The range of food is $\{27, 31\}$, the range of lid is $\{6, 13, 19, 21\}$, and the range of pot is $\{9, 17, 23, 29\}$.

**(D) Hang-Cup.** The range of the first cup is $\{4, 16, 22, 29\}$, the range of the second cup is $\{13, 14, 20, 27\}$, and the range of shelf is $\{11, 18\}$.

**(E) Stack-Brick.** The range of the first brick is $\{8, 9, 22, 23\}$, the range of the second brick is $\{13, 19, 21, 27\}$, and the range of the third brick is $\{10, 25\}$.

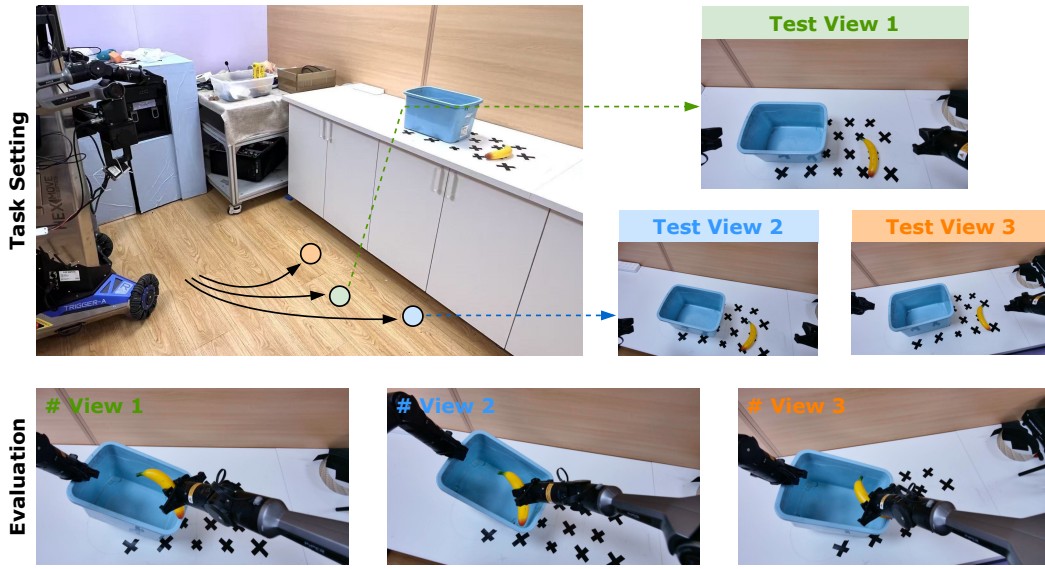

Figure 11: Visualization of mobile manipulation results. The policy trained with R2RGen successfully generalizes to different camera views with only one human-collected demonstration.

**(F) Build-Bridge.** The range of the first bridge pier is $\{8, 9, 22, 23\}$, the range of the second pier is $\{13, 14, 26, 27\}$, and the range of the bridge deck is $\{11, 24\}$.

The range of object locations of dual-arm tasks is shown in Figure 10 (b):

**(G) Grasp-Box.** The range of box is $\{1, 2, 3, 4, 5, 6, 7, 8, 10, 11, 12, 13, 20, 21, 22, 23, 24, 26, 27, 28, 29, 30, 31, 32\}$.

**(H) Store-Item.** The range of box is $\{1, 14, 27\}$, and the range of banana is $\{4, 6, 17, 19, 23, 26, 30, 32\}$.

Note the black crosses in Figure 10 indicate possibly seen locations during training, where human demonstrations are collected within these locations. The red crosses denote unseen locations which the training set does not cover.

## D    APPLICATION

R2RGen makes our 3D policy achieve strong spatial generalization across different viewpoints without camera calibration, so we can achieve mobile manipulation by simply combining a navigation system Wu et al. (2025) and a manipulation policy trained with R2RGen. Since the termination condition of navigation is relatively loose, the robot may stop at different docking point around the manipulation area, which imposes great challenges on the manipulation policy.

According to Figure 11, using iDP3 trained with R2RGen, the policy successfully generalizes to different docking points with maximum distance larger than 5cm. Different from DemoGen (DP3) which requires a careful calibration of the camera pose to crop environment pointclouds, our method directly applies on raw RGB-D observations during both data generation and policy training / inference stages, which is more practical in real-world applications.

## E    USAGE OF LLM

In the preparation of this manuscript, large language models (LLMs) were employed solely for the purpose of linguistic polishing and refinement of partial sentences. Specifically, LLMs were used to improve grammatical accuracy, enhance sentence fluency, and ensure terminological consistency in certain paragraphs. All conceptual development, theoretical analysis, experimental design, result interpretation, and methodological discussions remain entirely the work of the human authors. The

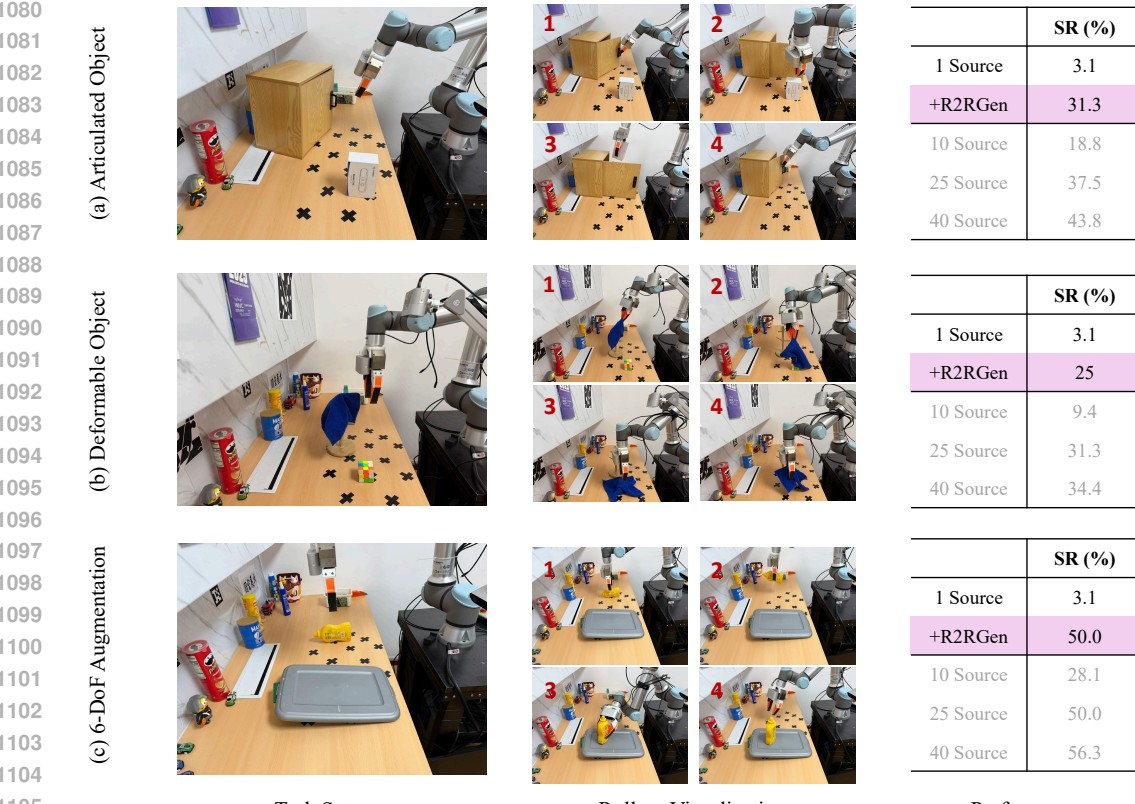

Figure 12: Apply R2RGen for articulated objects, non-rigid objects and 6-DoF augmentation.

use of LLMs did not contribute to the generation of original ideas, technical content, or scientific conclusions presented in this paper. All authors take full responsibility for the integrity and validity of the research and the content of this publication.

## F REBUTTAL MATERIAL

### F.1 EXPERIMENTS ON NON-RIGID OBJECTS AND 6-DOF AUGMENTATION

We provide additional experiments to apply R2RGen for articulated and deformable objects. We also validate that R2RGen supports 6-DoF object augmentation including pitch/roll changes.

**Articulated object.** By replacing FoundationPose with other 6-DoF pose estimation models supporting articulated object (e.g. ANCSH Li et al. (2020)), we can complete the pointclouds of articulated objects without any modification on R2RGen's pipeline. As shown in Figure 12 (a), we design an Store-Cabinet task. The agent is required to first open the door of a cabinet. Then it picks up an item and puts the item in the cabinet. Finally it close the door of cabinet. With only 1 source demonstration, the policy trained with R2RGen approaches the performance of one trained with 25 human demonstrations.

**Deformable object.** We can also replace FoundationPose with vision foundation models that support 3D pointcloud completion for deformable object (e.g. GarmentNets Chi & Song (2021)). However, current deformable object completion methods are often limited to specific object categories, which requires additional data collection and model training for novel objects. Given the time constraints during the rebuttal period, we opted to employ SAM2 for object point cloud tracking without explicit completion. To alleviate the visual mismatch issue caused by incomplete point clouds, we collected source demonstrations using a top-down camera view. This setup ensures that object augmentations introduce only minimal incomplete regions in the observations. As shown in Figure 12 (b), we design

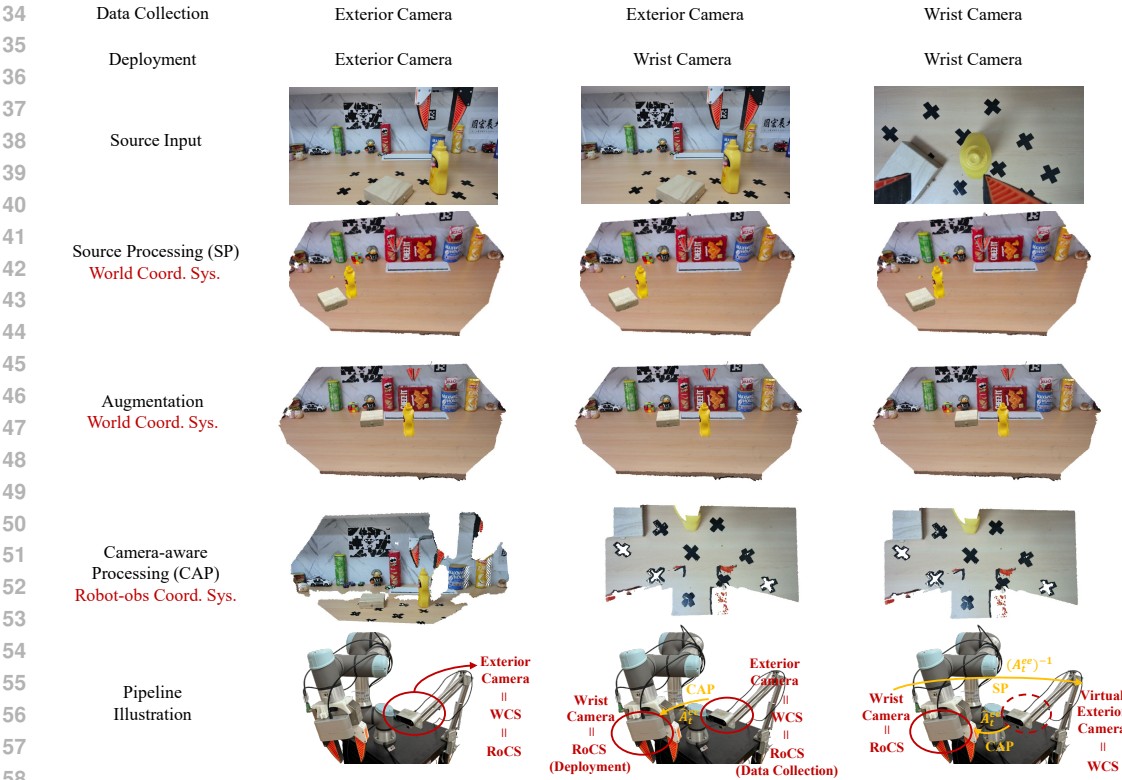

Figure 13: Extension of R2RGen to different camera settings.

an Cover-Cloth task. The agent is required to grasp a piece of cloth hanging on a rack, drape it over a Rubik's cube, lift the wrapped cube, and finally place it onto a platform. With 1 source demonstration, R2RGen successfully makes the policy generalize to various spatial configurations.

**6-DoF object augmentation.** We extend the Place-Bottle task to full 6-DoF object poses, as illustrated in Figure 12 (c). During evaluation, the bottle may be placed in arbitrary orientations beyond XY-plane translation and Z-axis rotation. To synthesize corresponding manipulation demonstrations, we apply 6-DoF augmentations to the bottle while ensuring that each resulting pose is physically plausible—respecting gravity constraints and preventing collisions between the bottle and the table surface. It is shown that with 1 source demonstration, R2RGen successfully makes the policy generalize to various 6-DOF object poses, which competes the policy trained with 25 human demonstrations.

### F.2 SUPPORT FOR WRIST CAMERA SETTING

In our main paper, we assume the camera is an exterior camera which is static during task execution for simplicity. But as a general framework, R2RGen also supports the widely adopted wrist camera setting without modifying the core logic.

**Definition of coordinate system.** We define the exterior camera frame as the world coordinate system (WCS) and align actions to this system. The frame of the equipped camera is referred to as the robot-observation coordinate system (RoCS), which may differ in data collection and deployment. Note that the action $\mathbf{A}_t^{ee}$—the SE(3) end-effector pose—naturally defines the transformation from the exterior camera frame to the wrist camera frame, since the wrist camera is rigidly attached to the end-effector. We show three typical settings in Figure 13, although our paper mainly focuses on the first setting (exterior camera for data collection and deployment), we will validate R2RGen also support other two settings (collect data with exterior or wrist camera, but only use wrist camera as the input sensor for manipulation policy). The relationship between different coordinate systems and cameras are illustrated in the figure.

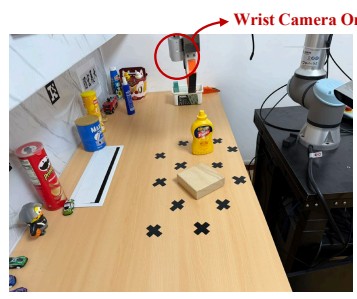 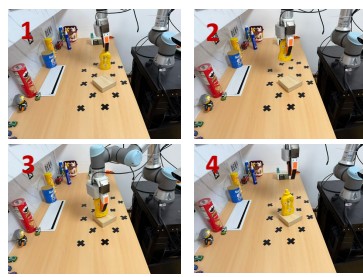

| | SR (%) |
|---|---|
| 1 Source | 3.1 |
| +R2RGen | 43.8 |
| 10 Source | 25 |
| 25 Source | 31.3 |
| 40 Source | 46.9 |

Task Setup          Rollout Visualization          Performance

Figure 14: Apply R2RGen for wrist camera setting.

**A unified formulation of R2RGen.** Recall R2RGen's three stage: source processing, group-wise data augmentation and camera-aware post-processing. The core augmentation part should modify both pointcloud observations and actions in WCS. Thus, the source processing stage should convert RoCS (of data collection) observation to WCS, while the post-processing stage converts WCS observation back to RoCS (of deployment). So a unified formulation of R2RGen for general camera configurations can be written as:

$$\{P_t^{SP}\} = \texttt{Scene-Parse}(\{P_t\}) \cdot \texttt{Transform}(\text{RoCS} \rightarrow \text{WCS}, \{(\mathbf{A}_t^{ee})^{-1}\}) \tag{8}$$

$$\{\hat{P}_t\}, \{\hat{\mathbf{A}}_t^{ee}\} = \texttt{Group-Augment}(\{P_t^{SP}\}, \{\mathbf{A}_t^{ee}\}) \tag{9}$$

$$\{\hat{P}_t^{adjust}\} = \texttt{Camera-Process}(\{\hat{P}_t\} \cdot \texttt{Transform}(\text{WCS} \rightarrow \text{RoCS}, \{\hat{\mathbf{A}}_t^{ee}\})) \tag{10}$$

Here $\texttt{Transform}(\cdot)$ will be an identity matrix if WCS equals to RoCS. Otherwise it leverages the original actions $\{(\mathbf{A}_t^{ee})^{-1}\}$ to convert the pointcloud observed by wrist camera to WCS, or uses the augmented actions $\{\hat{\mathbf{A}}_t^{ee}\}$ to convert the WCS pointcloud back to the wrist camera.

**Experiments on wrist camera setting.** We further conduct real-world experiments on the second setting shown in Figure 13, i.e., data collection with exterior camera while deployment with wrist camera. As shown in Figure 14, we reuse the Place-Bottle task in our main experiments while modifying the camera setting as described above. The experimental results validate the extension ability of R2RGen.

### F.3 STATISTICAL ANALYSIS OF EXPERIMENTAL RESULTS

**Failure case analysis.** To analyze the failure distribution, we classified error cases into six categories, as shown in Figure 15 (a). As requested by the reviewer, we provide the comparative failure matrix for the Open-Jar and Place-Bottle tasks between DemoGen and R2RGen. The results demonstrate that R2RGen improves the safety of the generated trajectories and the visual perception capability for better execution of the tasks.

**Statistical significance.** To evaluate the statistical significance of our results, we selected three representative main tasks: Pot-Food, Build-Bridge, and Grasp-Box. We calculated the 95% Wilson Score Intervals for success rates and performed Fisher's Exact Test against the 25x Human baseline, as shown in Figure 15 (b). R2RGen demonstrates a statistically improvement in the Pot-Food ($p = 0.0091$) and Grasp-Box ($p = 0.0360$) tasks. The statistical analysis confirms that in the evaluated scenarios, 1 Human Demo + R2RGen outperforms 25 Human Demos.

**Ablation on hyperparameter.** To evaluate the impact of key hyperparamters on the final performance of R2RGen, We performed an additional analysis varying the Patch-Z-Buffer radius ($r$) from 0 to 4 pixels, as shown in Figure 15 (c). The success rate peaks and remains stable across $r \in [1, 2]$ due to the best geometric fidelity.

### F.4 COMPARISON WITH SIMULATION-BASED METHODS

In addition to DemoGen, we further compare R2RGen with simulation-based methods including MimicGen Mandlekar et al. (2023), SplatSim Qureshi et al. (2025). Regarding GSWorld Jiang et al.

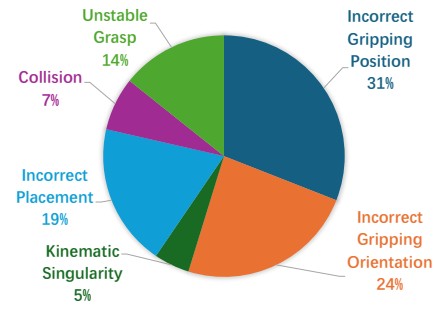

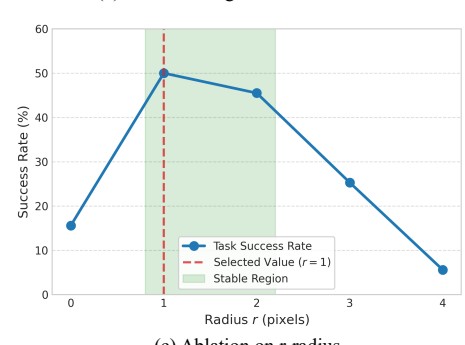

|  |  | **Success Rate** | **95% CI** | **P-value** |
|---|---|---|---|---|
| **Pot-Food** | **R2RGen** | 37.5 | [29.6%, 46.1%] | 0.0091 |
|  | 25x Human | 21.9 | [15.6%, 29.8%] |  |
| **Build-Bridge** | **R2RGen** | 34.4 | [26.7%, 43.0%] | 0.1525 |
|  | 25x Human | 28.1 | [21.1%, 36.5%] |  |
| **Grasp-Box** | **R2RGen** | 41.7 | [33.9%, 49.8%] | 0.0360 |
|  | 25x Human | 29.2 | [22.4%, 37.1%] |  |

(b) Statistical Significance

|  | R2RGen | Demogen |
|---|---|---|
| **Incorrect Gripping Position** | 45 | 80 |
| **Incorrect Gripping Orientation** | 35 | 53 |
| **Incorrect Placement** | 27 | 39 |
| **Kinematic Singularity** | 5 | 7 |
| **Unstable Grasp** | 20 | 35 |
| **Collision** | 10 | 24 |
| **Total Failures** | 144 | 238 |

(a) Failure Case

(c) Ablation on r-radius

Figure 15: Statistical analysis of experimental results

(2025) mentioned by the reviewer, we have currently omitted it from our comparison as the authors have not yet fully open-sourced the sim2real implementation, as indicated on their GitHub repository. We will include comparison in the final version if the code becomes available.

Comparative experiments of total data generation time and task success rates are conducted on Place-Bottle and Hang-Cup tasks, all methods using only 1 source demonstration for fair comparison. As shown in Table 5, R2RGen achieves better performance on these two tasks with faster speed. This mainly

Table 5: Comparative experiments of R2RGen and simulation-based methods. The generation time ($*$) is time cost for each trajectory. The rollout time ($\dagger$) of MimicGen includes 10 times of real-world rollouts.

|  | Success Rate (%) | | Time | | |
|---|---|---|---|---|---|
|  | Place-Bottle | Pot-Food | Setup | Generation$^*$ | Rollout$^\dagger$ |
| 1 Source | 3.1 | 3.1 | – | – | – |
| +MimicGen | 34.3 | 9.4 | 10min | 20s | 15min |
| +SplatSim | 40.6 | 34.4 | 40min | 25s | – |
| ***+R2RGen*** | **50.0** | **37.5** | **8min** | **1.5s** | – |

contributes to the real-to-real data generation paradigm, which bypasses both the real-to-sim and sim-to-real transitions. Skipping real-to-sim process makes R2RGen easy to use and fast, since simulation-based methods needs to build an accurate digital twin in simulator, which includes object / scene scanning, 3D Gaussian optimization and careful alignment of several coordinate systems. Moreover, being free of sim-to-real problem is the key to achieve high performance in real-world tasks. Although Gaussian-splatting–based simulation methods like SplatSim are also able to render photo-realistic images, the 3D Gaussian representation may be damaged with the moving of robotic arm and objects. Direct manipulation of Gaussian particles without finetuning Gaussian parameters often results in blurred rendering artifacts, particularly along modified region boundaries.

We will cite relevant papers Qureshi et al. (2025); Jiang et al. (2025); Yang et al. (2025) and add detailed discussion with them in the revised version of our paper.

