# OpenReview forum: "R2RGEN: Real-to-Real 3D Data Generation for Spatially Generalized Manipulation"
_ICLR.cc/2026/Conference — Submitted to ICLR 2026_

### Official Review · Reviewer_D69J · 2025-10-25

**Soundness:** 3
**Presentation:** 3
**Contribution:** 2
**Rating:** 6
**Confidence:** 5

**Summary:**

The paper introduces R2RGen, a simulator‑ and rendering‑free framework that generates large amounts of real‑to‑real 3D training data for robotic manipulation from as little as a single human demonstration. Instead of collecting many demonstrations to cover spatial variation (object layouts, robot base/viewpoint), R2RGen edits both the point‑cloud observations and the action trajectories directly in 3D, then trains a 3D visuomotor policy (iDP3) purely on the generated data.

Key components:

Pre‑processing: Parse a single demo into complete object, environment, and arm point clouds using segmentation + template‑based 3D tracking; lightly annotate skill/motion segments and the in‑hand/target objects

Group‑wise augmentation with backtracking: Treat each skill as acting on a group (targets + in‑hand), apply identical SE(3) transforms per group, and backtrack from the final skill while maintaining a set of “fixed” objects to preserve causal and multi‑object spatial constraints; plan motions to connect skills.



Overall, I think this paper is technically sound. However, the main concern lies in its novelty and contribution. Most parts of the system’s implementation and pipeline design appear quite straightforward, and I’m somewhat confused about the claimed level of generalization in this work.

**Strengths:**

Real‑to‑real pipeline that edits both observations and actions. Directly augments point‑cloud observations and end‑effector trajectories in a shared 3D frame; no on‑robot rollouts needed to regenerate visuals.

Handles multi‑object & bimanual skills. The group‑wise augmentation + backtracking idea preserves inter‑object structure (targets + in‑hand) and causality across skills—something object‑centric methods struggle with.

Camera‑aware post‑processing. Project → crop → patch‑wise Z‑buffer → fill → unproject aligns synthetic clouds with RGB‑D sensor statistics, mitigating the “visual mismatch” that plagues large transforms/viewpoint changes.

**Weaknesses:**

Rigid‑object & known‑template bias. The completion/tracking relies on template‑based 3D tracking and assumes rigidity; non‑rigid/unknown objects or significant shape variation are out of scope.

Environment prerequisites. Needs an empty‑scene capture to build a complete environment cloud and assumes scene static within a trial—less realistic for cluttered/dynamic settings.

Planar/task assumptions. Augmentations use XY translations + Z‑yaw with a tabletop plane fit; generalization to shelves, drawers, vertical workspaces, or tasks needing pitch/roll changes is unclear.

Feasibility checks are under‑specified. Random group transforms plus motion planning can create unreachable poses, collisions, or implausible contacts; the paper doesn’t detail strong kinematic/visibility rejection tests, so label noise risk remains.

Baseline coverage is narrow. Head‑to‑head comparisons focus on DemoGen and only on tasks it can handle; there’s no thorough comparison to simulator‑based generators (e.g., MimicGen variants with minimal rollouts) or strong 2D/3D data‑augmentation baselines.

It is apparent that the simulation generated data is cheaper than the real world operation. Thus, the rendering-free and simulation free setting is somewhat not make sense.

**Questions:**

The method appears straightforward to implement, and I’m not yet convinced about its core insight or novelty. The contribution currently reads as a relatively small idea. Could you provide a more precise, mathematically grounded formulation to clarify what is fundamentally new here?

My main concern is the “simulation- and rendering-free” claim. It seems that the same data generation could be accomplished in simulation with concise code—for example, using GSworld or other Gaussian-splatting–based densification pipelines. As a result, the argument against simulation feels weak. Please provide concrete evidence and justification for avoiding simulation as a data-generation source (e.g., quantitative comparisons, failure cases where simulation underperforms, or constraints that make simulation impractical for your setting).

GSWorld: Closed-Loop Photo-Realistic Simulation Suite for Robotic Manipulation

---

> ### Author Response · Authors · 2025-11-20
>
> Thank you for your valuable comments and kind words to our work. Below we address specific questions.
>
> **Weakness1: Assumption on rigid object and known template**
>
> Our framework is general and not limited to any specific object completion method. Currently, we employ FoundationPose for object parsing, which primarily handles rigid objects. To extend our method to articulated or deformable objects, we can directly replace it with other vision foundation models (such as ANCSH [1] or GarmentNets [2]) that are specifically designed for such object categories to complete the 3D geometry.
>
> Besides using specialized models for object parsing, we can also just utilize SAM2 for object pointcloud tracking without completion, as described in **Appendix A.1** (Parsing non-rigid objects). Although this may introduce visual mismatch problem, it is possible to generate diverse manipulation data with minimal visual mismatch by carefully setting the camera view. Both alternative approaches can be directly integrated into our pipeline without any modifications. Please refer to **Appendix F.1** for corresponding experimental results.
>
> Looking forward, with the emergence of more vision foundation models, we believe a unified model will eventually address 3D shape completion for non-rigid objects. For instance, a recent work SAM3D [3] demonstrates great potential to replace FoundationPose with support to both rigid and non-rigid objects. So current R2RGen can continue to benefit from future improvements of vision models without requiring any modifications to its core framework.
>
>
> **Weakness2: Environment prerequisite & How to apply for cluttered / dynamic environment**
>
> We appreciate the reviewer's insightful questions regarding environmental assumptions.
>
> Regarding the environment scan, our method does not require a perfectly clean scene. Our experimental environments already contained various background objects beyond the task-relevant ones. We simply remove the task-related objects for scanning, which is straightforward and brings negligible overhead during data collection.
>
> For dynamic or cluttered environments where non-task-related objects might move, our framework can naturally simulate such variations by applying data augmentation to the background point cloud during generation. The learned policy primarily focuses on the point cloud in the operational area and demonstrates relative robustness to environmental changes, as the task success depends mainly on the configuration of the target objects.
>
> **Weakness3: Planar / task assumption**
>
> While most of our experiments were conducted on horizontal surfaces for consistency with prior work, our method is not fundamentally limited to planar scenarios. The framework naturally supports augmentations with inclined planes and higher degrees of object freedom through appropriate geometric transformations. We have conducted additional experiments evaluating our approach on inclined surfaces and with more complex object augmentations, with results provided in **Appendix F.1**. These results demonstrate our method's capability to handle such variations effectively.
>
> **Weakness4: feasibility check**
>
> We thank the reviewer for raising this important point. We do perform feasibility checks in our pipeline. Specifically, after generating new demonstrations, we filter out infeasible data based on the robot arm's reachable workspace and prevent collisions between the robot arm (or in-hand objects) and the scene point cloud.
>
> For dual-arm manipulation tasks, in addition to implementing reachability and collision detection for both arms, we impose additional constraints to maintain action consistency for the in-hand object. These mechanisms are described in **Appendix A.3**.
>
> We will include more detailed implementation specifics regarding our feasibility checks in the revised version of the paper.
>
> **Weakness5: comparison with simulator-based baselines**
>
> Thank you for this suggestion. We have conducted additional comparisons with several representative simulator-based baselines. The experimental results are provided in **Appendix F.4**.

---

> > ### Author Response · Authors · 2025-11-20
> >
> > **Weakness6 & Question2: Real-to-real v.s. Simulation-based methods**
> >
> > We thank the reviewer for this insightful question regarding the comparison between real-to-real and simulation-based paradigms.
> >
> > We agree that simulation can provide a cheaper data source. However, it faces several inherent challenges: efficiency issues (constructing digital twins, loading simulation environments, and rendering), the sim-to-real generalization gap, and difficulties in simulating deformable objects.
> >
> > For a comparison, R2RGen's key advantage lies in bypassing both the real-to-sim and sim-to-real transitions. This makes our approach efficient, easy to use and completely free from transfer problems. While the primary strengths of simulation-based methods are more complete 2D/3D observations and more accurate robot dynamics modeling.
> > Note that although 3D Gaussian-based simulation methods can partially address the sim-to-real gap, they incur additional computational costs for scene-specific Gaussian optimization and still produce blurred renderings when Gaussian particles are manipulated without subsequent parameter fine-tuning.
> >
> > We believe both paradigms have complementary strengths and can benefit from each other's developments. We have added comparative experiments of total data generation time and task success rates in **Appendix F.4**. Furthermore, while simulation-based methods struggle with deformable objects, our framework can readily handle such tasks by replacing FoundationPose with other vision foundation models, as demonstrated in **Appendix F.1**.
> >
> > **Question1: Core insight or novelty**
> >
> > We thank the reviewer for the opportunity to clarify our core technical contributions. The primary contribution of this work is a comprehensive real-to-real data generation framework that efficiently handles arbitrary numbers of objects and diverse interaction modes, enabling rapid real-world policy training and deployment without requiring simulators.
> >
> > Our key technical innovations are:
> >
> > (1) **A Novel Three-Stage Pipeline**: We propose a systematic approach comprising:
> >
> > Pre-processing: Decoupling objects / scene and skill / motion in a shared world coordinate system
> >
> > Augmentation: Coordinated point-cloud and action augmentation in world coordinates
> >
> > Post-processing: Camera-Aware Processing to align generated data with real sensor observations and coordinate system
> >
> > This pipeline is simple, general, and adaptable to various settings. For non-rigid objects, FoundationPose can be directly replaced with SAM2 without altering the pipeline logic. Similarly, for different sensor configurations, we can apply the same unified approach through coordinate transformations. For instance, in settings like a wrist-mounted camera, observations and actions can first be transformed to a world coordinate system for augmentation, and then converted back to the camera frame in a post-processing step. Detailed illustrations and experimental results supporting this extension are provided in **Appendix F.2**.
> >
> > (2) **Backtracking Group-wise Augmentation**: We introduce a novel backtracking mechanism and group-wise augmentation strategy that dynamically maintains spatial constraints among multiple objects. This fundamental advancement enables handling arbitrary object counts and complex interaction patterns—a significant limitation of prior methods like DemoGen.
> >
> > These technical contributions are validated through extensive experiments in multi-object tasks, demonstrating superior performance over existing methods while maintaining remarkable simplicity and efficiency.
> >
> > [1] Category-Level Articulated Object Pose Estimation. CVPR 2020.
> >
> > [2] GarmentNets: Category-Level Pose Estimation for Garments via Canonical Space Shape Completion. ICCV 2021.
> >
> > [3] SAM 3D: 3Dfy Anything in Images. 2025.
> >
> >
> > **If you have any further question, please let us know. Thank you very much!**

---

> ### Comment · Reviewer_D69J · 2025-11-20
>
> Thanks for the authors' feedback, i think most of my concern and worries has been addressed.
>
> I will maintain my score of weak accept.
>
> Also for the statement that " Furthermore, while simulation-based methods struggle with deformable objects, our framework can readily handle such tasks by replacing FoundationPose with other vision foundation models". I just want to remind that some of the recent work has solved this issue, like Real-to-Sim Robot Policy Evaluation with Gaussian Splatting Simulation of Soft-Body Interactions.

---

> ### Author Response · Authors · 2025-11-21
>
> Dear reviewer:
>
> We sincerely thank you for your positive feedback and for maintaining your score. We fully agree that Gaussian-based simulation methods represent a promising direction, and we appreciate you pointing us to relevant advances in simulating deformable objects. Indeed, as new techniques like the Gaussian splatting approach you mentioned continue to mature, they are likely to overcome current limitations.
>
> At the same time, R2RGen is designed as a modular framework that can continually benefit from progress in vision foundation models—such as recent SAM 3D—without architectural changes. We view real-to-real and simulation-based methods as complementary routes forward, each with distinct strengths. Our work offers an alternative pathway, but does not diminish the value of simulation-based pipelines. We look forward to seeing how different technical streams evolve, cross-pollinate, and eventually converge in future work.
>
> We will continue to polish the presentation and are committed to releasing the code to facilitate further research in this direction. Thank you again for your thoughtful input :)

---

### Official Review · Reviewer_AThM · 2025-10-28

**Soundness:** 4
**Presentation:** 4
**Contribution:** 4
**Rating:** 6
**Confidence:** 4

**Summary:**

R2RGen offers a real enhancement Pipeline based solely on visible light video, which can reliably scale a single person's demonstration to hundreds of different point cloud trajectories. It achieves a success rate comparable to that of using 25 times the human demonstration trajectory on 8 real robot tasks with just one human demonstration trajectory +R2RGen. This work has been meticulously designed and extensively evaluated, demonstrating excellent deployment value. However, some design choices lack quantitative sensitivity analysis, and the experiments did not further explore the statistical significance of the results and failure cases. Moreover, under basis uncertainty, the movement operation expansion cannot maintain external calibration - although the movement operation is only used as a test case for limit generalization testing and is mainly left for further consideration in the future. But overall, this is a very valuable piece of work.

**Strengths:**

1.In response to the problem that previous works could not handle the spatial changes of multi-object structures well, an original "group-level retroactive augmentation" method was proposed to maintain the constraints of multi-object structures, breaking through the single-object limitation of the Baseline method DemoGen.
2. The camera perception post-processing proposed by R2RGen addresses occlusion/missing caused by large rotations, significantly reducing visual mismatch.
3. The experimental scale is sufficient and the experimental process is relatively complete: approximately 1,500 real-machine rollout operations, 8 different types of grasping operation tasks, and it simultaneously includes 4 different levels of ablation experiments, with high confidence in the results.

4.The entire process is based on real-world RGB video data, without the need for simulation or rendering. The mobile base is plug-and-play, and the engineering value is clear.

**Weaknesses:**

1.Zero analysis of failure cases: Although the experimental volume of the paper is very sufficient and the improvement in the Success Rate is significant compared to the baseline method DemoGen, the analysis of the experimental results only provides the success rate and does not classify failure results such as collision, non-capture, and out-of-bounds, nor does it conduct further analysis of failure cases.
2. Insufficient statistical significance of the experimental results: The main results of the paper, Table 1, lack confidence intervals or hypothesis tests. Relying solely on the current data to illustrate that "achieving the effect of using 25 times the manual demonstration trajectory" may not be the most rigorous.
3. Some hyperparameter sensitive data in the experiment are missing. For instance, the patch-Z-buffer radius r and the depth threshold δ are only given as single values. These two parameters obviously affect the occlusion judgment and the final point cloud quality, but there is no sensitivity curve for the experimental hyperparameters.
4. Although the movement operation is only used for extreme generalization tests, the assumption of using a fixed camera leads to loopholes in the experimental process of the movement operation: assuming that "the RGB-D camera is fixed to the base", the navigation stop error in the movement operation experiment (§ 4.5&D) is greater than 5 cm, which causes the actual external parameters of the camera to change and may introduce geometric inconsistencies.

**Questions:**

1.In Table 1, all success rates only provide point estimates for a single round of 32 to 64 rolluts, with neither standard deviations nor 95% confidence intervals. For success rates at the 3% to 50% level, when n≤100, the half-width of the binomial interval can reach ± 10%, making it impossible to determine whether "R2RGen is superior to 25× human" is significant. Please provide the 95% interval of the success rate of the main tasks and the significance test results compared with 25 source-human-video.
2. In the main text, only "success rate" is clearly stated, but the failure cases (collision, non-contact, target placement beyond the limit, unfeasible trajectory, etc.) are not classified. The generated data may systematically magnify certain types of errors (such as placement deviations after large rotations), but readers cannot determine whether such problems exist merely based on the main text content. Please provide the confusion matrix of failure cases and compare the failure distribution of DemoGen and R2RGen to explain whether the generation strategy reduces specific errors.
3. In the mobile operation experiment, the camera post-processing still used the same set of internal and external parameters, and no methods such as online reprojection to reduce the impact were performed, which might introduce geometric inconsistencies. If possible, could the distribution of navigation stop errors (mean ±std) be provided in the future?
4. The generated data uses a high-density completed point cloud, while the input during deployment is a 640×480 original depth downsample. Can this supplement the explanation for "inconsistent training-test resolutions" and verify the robustness of the strategy to density differences?
5. The experiments mentioned in the section "Relationship between Performance and Annotation" of the paper were conducted on four types of sources: 1, 3, 5, and 10. However, the figures shown in Figure 5 are for the four types of sources: 1, 2, 3, and 5. Please check, proofread, and make corrections.

---

> ### Author Response · Authors · 2025-11-20
>
> Thank you for your valuable comments and kind words to our work. Below we address specific questions.
>
> **Weakness1 & Question2: Failure case analysis**
>
> We thank the reviewer for this valuable suggestion. We have actually presented some typical failure cases in Figure 7 and our project page (including video demonstrations). Following your suggestion, we have conducted additional experiments to provide a more comprehensive failure analysis, which can be found in **Appendix F.3**.
>
> **Weakness2 & Question1: Statistical significance**
>
> Following the reviewer's suggestion, we provide more details about statistical significance of the experimental results in **Appendix F.3**.
>
> **Weakness3: Ablation on hyperparameters**
>
> Following the reviewer's suggestion, we provide ablation experiments on some key hyperparameters, as shown in **Appendix F.3**.
>
> **Weakness4 & Question3: About movement operations in mobile manipulation**
>
> In our mobile manipulation experiments, the robot first moves using an existing navigation method, and then executes fixed-base manipulation using the iDP3 policy trained with R2RGen-generated data. Since iDP3 requires only an egocentric point cloud as input, the policy execution does not require camera extrinsics during deployment.
>
> We clarify that camera extrinsics are only needed during the data generation phase to ensure proper object and environment point cloud augmentation in world coordinates. Once the data is generated and the policy is trained, the deployed iDP3 model operates solely using egocentric point cloud observations, enabling straightforward real-world deployment without requiring additional extrinsic calibration.
>
> **Question4: Inconsistent training-test resolutions**
>
> We appreciate the reviewer's attention to this implementation detail. We confirm that the point cloud resolutions used during training and testing are identical, as both are derived from RGB-D images using the same process.
>
> Furthermore, thanks to our camera-aware processing module, the point density and distribution in our generated data closely match those observed from real-world RGB-D sensors, ensuring consistency between the training data and real-world testing conditions.
>
> **Question5: Typos in Section 4.3**
>
> We thank the reviewer for pointing out this issue. We actually used 1, 2, 3 and 5 human demonstrations respectively, as correctly shown in Figure 5. We have corrected this typo in the main text accordingly.
>
>
> **If you have any further question, please let us know. Thank you very much!**

---

> > ### Comment · Reviewer_AThM · 2025-11-26
> >
> > The authors' feedback has satisfactorily addressed my primary concerns. I am maintaining my score of Weak Accept.

---

### Official Review · Reviewer_K8uL · 2025-10-29

**Soundness:** 4
**Presentation:** 3
**Contribution:** 3
**Rating:** 6
**Confidence:** 4

**Summary:**

R2RGen offers a real enhancement Pipeline based solely on visible light video, which can reliably scale a single person's demonstration to hundreds of different point cloud trajectories. It achieves a success rate comparable to that of using 25 times the human demonstration trajectory on 8 real robot tasks with just one human demonstration trajectory +R2RGen. This work has been meticulously designed and extensively evaluated, demonstrating excellent deployment value. However, some design choices lack quantitative sensitivity analysis, and the experiments did not further explore the statistical significance of the results and failure cases. Moreover, under basis uncertainty, the movement operation expansion cannot maintain external calibration - although the movement operation is only used as a test case for limit generalization testing and is mainly left for further consideration in the future. But overall, this is a very valuable piece of work.

**Strengths:**

- In response to the problem that previous works could not handle the spatial changes of multi-object structures well, an original "group-level retroactive augmentation" method was proposed to maintain the constraints of multi-object structures, breaking through the single-object limitation of the Baseline method DemoGen.

- The camera perception post-processing proposed by R2RGen addresses occlusion/missing caused by large rotations, significantly reducing visual mismatch.

- The experimental scale is sufficient and the experimental process is relatively complete: approximately 1,500 real-machine rollout operations, 8 different types of grasping operation tasks, and it simultaneously includes 4 different levels of ablation experiments, with high confidence in the results.

- The entire process is based on real-world RGB video data, without the need for simulation or rendering. The mobile base is plug-and-play, and the engineering value is clear.

**Weaknesses:**

- Zero analysis of failure cases: Although the experimental volume of the paper is very sufficient and the improvement in the Success Rate is significant compared to the baseline method DemoGen, the analysis of the experimental results only provides the success rate and does not classify failure results such as collision, non-capture, and out-of-bounds, nor does it conduct further analysis of failure cases.

- Insufficient statistical significance of the experimental results: The main results of the paper, Table 1, lack confidence intervals or hypothesis tests. Relying solely on the current data to illustrate that "achieving the effect of using 25 times the manual demonstration trajectory" may not be the most rigorous.

- Some hyperparameter sensitive data in the experiment are missing. For instance, the patch-Z-buffer radius r and the depth threshold δ are only given as single values. These two parameters obviously affect the occlusion judgment and the final point cloud quality, but there is no sensitivity curve for the experimental hyperparameters.

- Although the movement operation is only used for extreme generalization tests, the assumption of using a fixed camera leads to loopholes in the experimental process of the movement operation: assuming that "the RGB-D camera is fixed to the base", the navigation stop error in the movement operation experiment (§ 4.5&D) is greater than 5 cm, which causes the actual external parameters of the camera to change and may introduce geometric inconsistencies.

**Questions:**

1. In Table 1, all success rates only provide point estimates for a single round of 32 to 64 rolluts, with neither standard deviations nor 95% confidence intervals. For success rates at the 3% to 50% level, when n≤100, the half-width of the binomial interval can reach ± 10%, making it impossible to determine whether "R2RGen is superior to 25× human" is significant. Please provide the 95% interval of the success rate of the main tasks and the significance test results compared with 25 source-human-video.

2. In the main text, only "success rate" is clearly stated, but the failure cases (collision, non-contact, target placement beyond the limit, unfeasible trajectory, etc.) are not classified. The generated data may systematically magnify certain types of errors (such as placement deviations after large rotations), but readers cannot determine whether such problems exist merely based on the main text content. Please provide the confusion matrix of failure cases and compare the failure distribution of DemoGen and R2RGen to explain whether the generation strategy reduces specific errors.

3. In the mobile operation experiment, the camera post-processing still used the same set of internal and external parameters, and no methods such as online reprojection to reduce the impact were performed, which might introduce geometric inconsistencies. If possible, could the distribution of navigation stop errors (mean ±std) be provided in the future?

4. The generated data uses a high-density completed point cloud, while the input during deployment is a 640×480 original depth downsample. Can this supplement the explanation for "inconsistent training-test resolutions" and verify the robustness of the strategy to density differences?

5. The experiments mentioned in the section "Relationship between Performance and Annotation" of the paper were conducted on four types of sources: 1, 3, 5, and 10. However, the figures shown in Figure 5 are for the four types of sources: 1, 2, 3, and 5. Please check, proofread, and make corrections.

---

> ### Author Response · Authors · 2025-11-20
>
> Thank you for your valuable comments and kind words to our work. Below we address specific questions.
>
> **Weakness1 & Question2: Failure case analysis**
>
> We thank the reviewer for this valuable suggestion. We have actually presented some typical failure cases in Figure 7 and our project page (including video demonstrations). Following your suggestion, we have conducted additional experiments to provide a more comprehensive failure analysis, which can be found in **Appendix F.3**.
>
> **Weakness2 & Question1: Statistical significance**
>
> Following the reviewer's suggestion, we provide more details about statistical significance of the experimental results in **Appendix F.3**.
>
> **Weakness3: Ablation on hyperparameters**
>
> Following the reviewer's suggestion, we provide ablation experiments on some key hyperparameters, as shown in **Appendix F.3**.
>
> **Weakness4 & Question3: About movement operations in mobile manipulation**
>
> In our mobile manipulation experiments, the robot first moves using an existing navigation method, and then executes fixed-base manipulation using the iDP3 policy trained with R2RGen-generated data. Since iDP3 requires only an egocentric point cloud as input, the policy execution does not require camera extrinsics during deployment.
>
> We clarify that camera extrinsics are only needed during the data generation phase to ensure proper object and environment point cloud augmentation in world coordinates. Once the data is generated and the policy is trained, the deployed iDP3 model operates solely using egocentric point cloud observations, enabling straightforward real-world deployment without requiring additional extrinsic calibration.
>
> **Question4: Inconsistent training-test resolutions**
>
> We appreciate the reviewer's attention to this implementation detail. We confirm that the point cloud resolutions used during training and testing are identical, as both are derived from RGB-D images using the same process.
>
> Furthermore, thanks to our camera-aware processing module, the point density and distribution in our generated data closely match those observed from real-world RGB-D sensors, ensuring consistency between the training data and real-world testing conditions.
>
> **Question5: Typos in Section 4.3**
>
> We thank the reviewer for pointing out this issue. We actually used 1, 2, 3 and 5 human demonstrations respectively, as correctly shown in Figure 5. We have corrected this typo in the main text accordingly.
>
>
> **If you have any further question, please let us know. Thank you very much!**

---

> > ### Author Response · Authors · 2025-11-27
> >
> > Dear reviewer:
> >
> > Since the discussion stage is nearing its end, we would appreciate your feedback and are happy to address any concerns you may have.

---

### Official Review · Reviewer_A5Dr · 2025-11-01

**Soundness:** 2
**Presentation:** 2
**Contribution:** 2
**Rating:** 4
**Confidence:** 2

**Summary:**

This paper proposes a method for training policies from a single real-world demonstration by generating additional training data directly in 3D point cloud space. To achieve this, the RGB-D observation of the demonstration is first processed into object and background point clouds. Then, the demonstration trajectory is segmented into individual skills. For each skill, the method transforms the relevant objects and robot end-effector poses together to create new variations of the same skill. Based on the augmented trajectories, new RGB-D observations from novel viewpoints can be rendered using the processed point clouds. A visuomotor policy is then trained using these synthetic trajectories and observations so that the robot can perform the task in new positions and environments. The approach is tested on real robots for multi-step manipulation tasks, showing that it improves generalization compared to using only the original demonstration and a baseline method.

**Strengths:**

- The paper is well-written and easy to read.
- The authors provide extensive experiments to demonstrate the effectiveness of the proposed method.

**Weaknesses:**

- A major weakness of this paper is its reliance on complete object geometry, which is a very strong assumption. In Supplementary A.1, the authors explicitly state that*each object is pre-scanned using an RGB-D camera to obtain the 3D mesh, which is then used for point cloud completion during data augmentation and policy learning. However, this introduces multiple concerns: First, this limits the method to rigid objects, because complete geometry of non-rigid objects cannot be easily achieved during task execution. Although the authors mention that they can fall back to incomplete point clouds for non-rigid objects, Table 2 shows that removing point cloud completion drops the success rate significantly, worse than the baseline method. Second, If objects are scanned, why not the environment?  With a scanned environment, one could render consistent background point clouds from arbitrary viewpoints as in [1].
- The assumption of a static robot base and a static camera during execution also limits the applicability of the proposed method.
- The overall technical contribution is limited. Although generalizing trajectory augmentation to interactions with more than two objects is interesting, scene parsing, trajectory parsing and camera-aware processing are all leveraging established techniques.



[1] Novel Demonstration Generation with Gaussian Splatting Enables Robust One-Shot Manipulation. Sizhe Yang, et al. RSS 2025.

**Questions:**

I believe the paper can be improved by lifting the known object geometry constraint.

---

> ### Author Response · Authors · 2025-11-20
>
> Thank you for your valuable comments. Below we address specific questions.
>
> **Weakness1-1: Reliance on complete object geometry**
>
> Our framework is general and not limited to any specific object completion method. Currently, we employ FoundationPose for object parsing, which primarily handles rigid objects. To extend our method to articulated or deformable objects, we can directly replace it with other vision foundation models (such as ANCSH [1] or GarmentNets [2]) that are specifically designed for such object categories to complete the 3D geometry.
>
> Besides using specialized models for object parsing, we can also just utilize SAM2 for object pointcloud tracking without completion. Although this may introduce visual mismatch problem, it is possible to generate diverse manipulation data with minimal visual mismatch by carefully setting the camera view. Both alternative approaches can be directly integrated into our pipeline without any modifications. Please refer to **Appendix F.1** for corresponding experimental results.
>
> Looking forward, with the emergence of more vision foundation models, we believe a unified model will eventually address 3D shape completion for non-rigid objects. For instance, a recent work SAM3D [3] demonstrates great potential to replace FoundationPose with support to both rigid and non-rigid objects. So current R2RGen can continue to benefit from future improvements of vision models without requiring any modifications to its core framework.
>
>
> **Weakness1-2: Performance drop when removing pointcloud completion**
>
> Regarding the performance drop when removing pointcloud completion in Table 2, we should clarify that this ablation is conducted under large spatial augmentation (same with original R2RGen to isolate the effect of shape completion) during data generation, whereas baseline methods (DemoGen) use smaller augmentation and manually crop background points to reduce training difficulty. If we also use smaller spatial augmentation to avoid severe visual mismatch caused by incomplete shape (while still keeping the raw RGB-D input without background cropping), the results are as follows:
>
> | Method | Place-Bottle | Hang-Cup |
> | :--------: | :---------: | :---------: |
> | DemoGen | 15.6 | Fail |
> | R2RGen w/o completion | 18.8 | 15.6 |
>
>
> The results show that even without completion, our method under smaller augmentation during data generation still competes the baseline, despite us using the raw RGB-D observation without background cropping. This demonstrates the robustness of our approach.
>
> **Weakness1-3: Why not scan the environment**
>
> We appreciate the reviewer's comment. In our implementation, we do scan the environment. Specifically, before data collection, we temporarily remove the manipulable objects and scan the static background to obtain a complete environment point cloud. This ensures that the background remains intact and free of holes when objects are rearranged during data generation. This procedure is described in the second paragraph of Section 3.2 of our paper.
>
> **Weakness2: Assumption on static base and camera**
>
> We thank the reviewer for raising this point. The assumption of a fixed base and camera during task execution is indeed a common setting in related work, adopted by all previous methods from MimicGen to DemoGen. Compared to these works, ours is the first to successfully validate effectiveness on mobile manipulation task, where we evaluate policies across diverse base positions achieved by navigation method. Experiments are shown in **Appendix D**.
>
> Furthermore, our framework is general and can be extended to support a moving base or camera. For instance, in settings like a wrist-mounted camera, observations and actions can first be transformed to a world coordinate system for augmentation, and then converted back to the camera frame in a post-processing step. Detailed illustrations and experimental results supporting this extension are provided in **Appendix F.2**.
>
> **To be continued in next reply.**

---

> > ### Author Response · Authors · 2025-11-20
> >
> > **Weakness3: Technical contribution**
> >
> > We thank the reviewer for the opportunity to clarify our core technical contributions. The primary contribution of this work is a comprehensive real-to-real data generation framework that efficiently handles arbitrary numbers of objects and diverse interaction modes, enabling rapid real-world policy training and deployment without requiring simulators.
> >
> > Our key technical innovations are:
> >
> > (1) **A Novel Three-Stage Pipeline**: We propose a systematic approach comprising:
> >
> > Pre-processing: Decoupling objects / scene and skill / motion in a shared world coordinate system
> >
> > Augmentation: Coordinated point-cloud and action augmentation in world coordinates
> >
> > Post-processing: Camera-Aware Processing to align generated data with real sensor observations and coordinate system
> >
> > This pipeline is simple, general, and adaptable to various settings. For non-rigid objects, FoundationPose can be directly replaced with SAM2 without altering the pipeline logic. Similarly, for different sensor configurations (e.g., wrist cameras), we can apply the same unified approach through coordinate transformations as discussed in **Appendix F.2**.
> >
> > (2) **Backtracking Group-wise Augmentation**: We introduce a novel backtracking mechanism and group-wise augmentation strategy that dynamically maintains spatial constraints among multiple objects. This fundamental advancement enables handling arbitrary object counts and complex interaction patterns—a significant limitation of prior methods like DemoGen.
> >
> > These technical contributions are validated through extensive experiments in multi-object tasks, demonstrating superior performance over existing methods while maintaining remarkable simplicity and efficiency.
> >
> > [1] Category-Level Articulated Object Pose Estimation. CVPR 2020.
> >
> > [2] GarmentNets: Category-Level Pose Estimation for Garments via Canonical Space Shape Completion. ICCV 2021.
> >
> > [3] SAM 3D: 3Dfy Anything in Images. 2025.
> >
> > **If you have any further question, please let us know. Thank you very much!**

---

> > > ### Author Response · Authors · 2025-11-27
> > >
> > > Dear reviewer:
> > >
> > > Since the discussion stage is nearing its end, we would appreciate your feedback and are happy to address any concerns you may have.

---

### Author Response · Authors · 2025-11-20
**Global response to reviewers**

Thanks for all reviewers’ constructive comments and patience on our work. We have provided detailed replies to the questions of each reviewer and refined our paper following the suggestions. The modification on our paper mainly including:

-	Experiments on non-rigid objects and 6-DoF augmentation in **Appendix F.1**.
-	Extension for wrist camera setting in **Appendix F.2**.
-	Statistical analysis of experimental results in **Appendix F.3**.
-	Comparison with simulation-based methods results in **Appendix F.4**.
-	Some writing and presentation are improved.

We will further refine the paper to properly incorporate all additional experiments and analyses conducted during the rebuttal period.

---

### Meta-Review · Area_Chair_1Hd5 · 2025-12-22

**Summary:**

The submission introduces a real-to-real 3D data generation framework that augments observation-action pairs to generate real-world data for robotic manipulation. Reviewers are concerned about the strong assumption (especially on the vision model), the limited scope, and insufficient technical contributions.

**Reviewer Concerns:**

While some other concerns may have been addressed, the most significant concern about the assumption of complete object geometry remains unaddressed. The AC disagrees with the authors that replacing with other vision foundation models will allow the work to be extended to articulated or deformable objects, as shape estimation of those objects is not well solved by any foundation models.

**Reviewer Scores:**

Reviewers will likely maintain their scores of 4, 6, 6, 6.  However, R2 and R3 have the same reviews, and only one will be considered.

---

### Decision · Program_Chairs · 2026-01-26

Reject